# Recognition and cleavage mechanism of intron-containing pre-tRNA by human TSEN endonuclease complex

Ling Yuan[1], Yaoyao Han [2], Jiazheng Zhao[1], Yixiao Zhang [2] ✉ & Yadong Sun [1] ✉

Removal of introns from transfer RNA precursors (pre-tRNAs) occurs in all living organisms. This is a vital phase in the maturation and functionality of tRNA. Here we present a 3.2 Å-resolution cryo-EM structure of an active human tRNA splicing endonuclease complex bound to an intron-containing pre-tRNA. TSEN54, along with the unique regions of TSEN34 and TSEN2, cooperatively recognizes the mature body of pre-tRNA and guides the anticodon-intron stem to the correct position for splicing. We capture the moment when the endonucleases are poised for cleavage, illuminating the molecular mechanism for both 3′ and 5′ cleavage reactions. Two insertion loops from TSEN54 and TSEN2 cover the 3′ and 5′ splice sites, respectively, trapping the scissile phosphate in the center of the catalytic triad of residues. Our findings reveal the molecular mechanism for eukaryotic pre-tRNA recognition and cleavage, as well as the evolutionary relationship between archaeal and eukaryotic TSENs.

Transfer RNA (tRNA) is an essential component in protein synthesis. It is initially transcribed as a precursor and must undergo a series of essential processing steps to form a mature, functional tRNA[1,2]. One of the key steps in this processing is the removal of introns. tRNA splicing occurs in all living organisms[3,4]. In particular, several isodecoders in humans, e.g., tRNA$^{Tyr}_{GTA}$, tRNA$^{Ile}_{TAT}$, and tRNA$^{Leu}_{CAA}$, are only encoded as intron-containing precursors[5–7], indicating the pivotal role of tRNA splicing in translation and cellular homeostasis. In eukaryotes, tRNA introns, ranging in length from 6 to 133 nucleotides (nts) are strictly located at the canonical position between nucleotides 37 and 38 of the mature tRNA, whereas intron positions in archaeal tRNAs are variable[8–10]. The removal of these introns is carried out by a conserved splicing machinery known as tRNA splicing endonuclease (EndA or TSEN). In archaea, four distinct types of EndAs that share similar architectural arrangements but employ different domains and subunit compositions are observed, including $\alpha_4$, $\alpha'_2$, $(\alpha\beta)_2$, and $\varepsilon_2$-type complexes[11,12], while the eukaryotic TSEN complex is a heterotetrameric complex consisting of four subunits: two structural units, TSEN15 and TSEN54, and two catalytic units, TSEN34 and TSEN2 (Fig. 1a)[13–16]. After cleavage, the two exon products, with a 5′-hydroxyl

terminus and a 2′,3′-cyclic phosphate terminus, are joined together by RNA ligase[17–19].

Structural studies on the EndA endonucleases from *Archaeoglobus fulgidus* have revealed that the cross-subunit cation-π-sandwich and a bulge-helix-bulge (BHB) RNA architecture are key features for substrate recognition and cleavage[20]. The splice sites, where the cleavage occurs, are located in the two three-nucleotide bulges, separated by a four-base pair helix[21]. The cross-subunit cation-π-sandwich means that the bugle is cleaved by one catalytic unit, but its first nucleobase is recognized by two arginine residues from another catalytic unit. This mechanism is well-suited for identifying the introns in archaea. Recent biochemical and structural studies have shown that human TSEN34 and TSEN15 form a heterodimer comparable to archaeal tRNA endonucleases[14]. However, earlier studies on the yeast Sen complex have shown that this cation-π mechanism is only required for catalyzing the 5′ splice site[22]. In humans, the TSEN complex specifically recognizes the mature body of pre-tRNA, with the so-called anticodon-intron (A-I) base pair playing a vital role in splicing (Fig. 1b, c)[14,23]. Intriguingly, TSEN54 is hypothesized to serve as a molecular ruler, defining the intron-exon junctions to ensure accurate

[1]School of Life Science and Technology, ShanghaiTech University, Shanghai, China. [2]Interdisciplinary Research Center on Biology and Chemistry, Shanghai Institute of Organic Chemistry, Chinese Academy of Sciences, Shanghai, China. ✉e-mail: yzhang@mail.sioc.ac.cn; sunyd1@shanghaitech.edu.cn

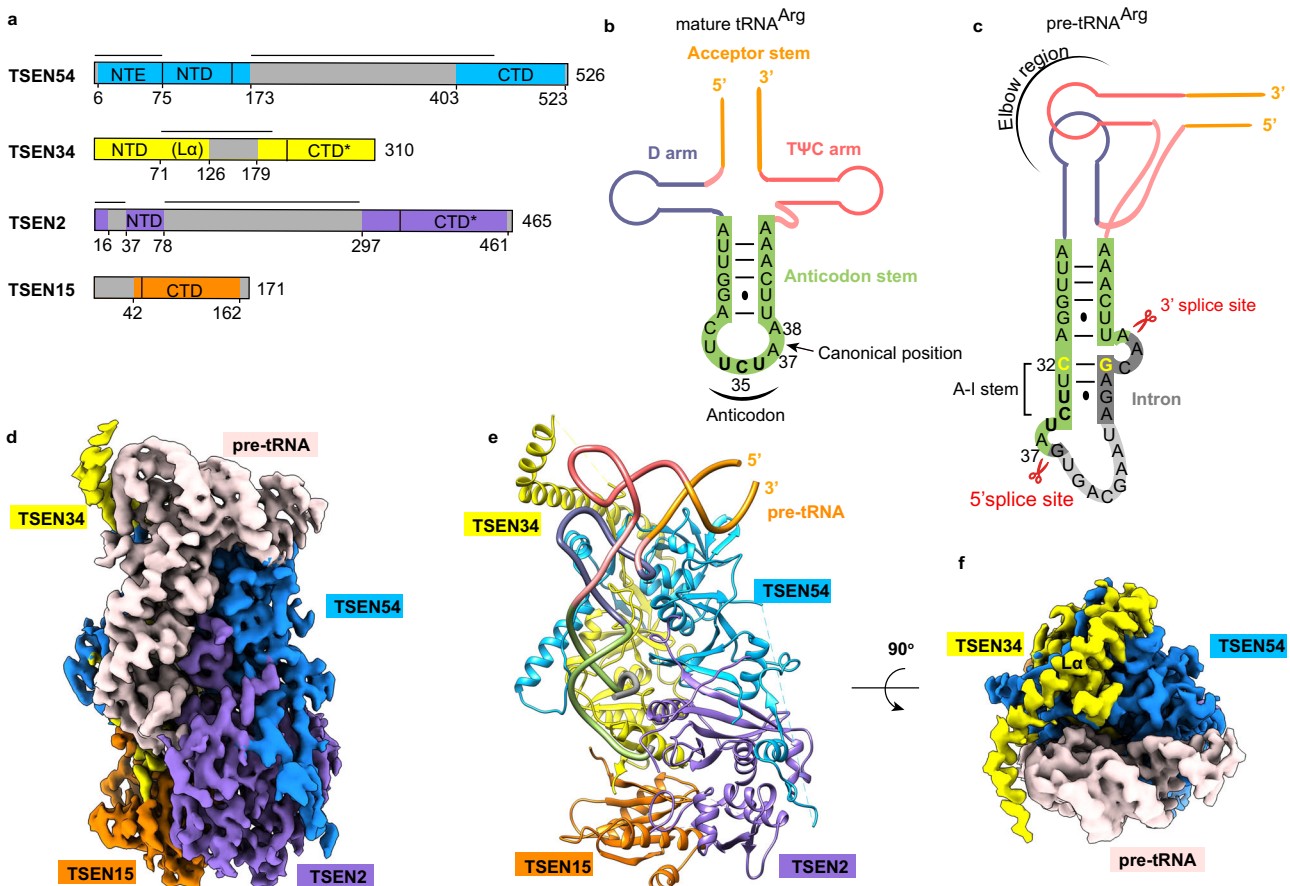

**Fig. 1 | Overall structure of the wild-type human TSEN–pre-tRNA^Arg complex.**
**a** Domain organizations of human TSEN54, TSEN34, TSEN2, and TSEN15. Domains having structural information from this study are shown as colored domains. Disordered segments are shown in gray, and catalytic domains are labeled with a '*' (star). The black lines represent the novel insertions in the human TSEN subunits as compared to the archaeal endonuclease. NTE, N-terminal extension; NTD, N-terminal domain; CTD, C-terminal domain; Lα, long α helix. **b** Schematic drawing of the secondary structure of mature tRNA^Arg from humans. Key regions of the tRNA are labeled. The anticodon is highlighted in bold. The intron insertion position is indicated by the black arrow. **c** Schematic representation of the folding of the pre-tRNA^Arg used in this study, with the same color scheme as in panel (**b**). The anticodon domain sequence is shown in green, and the intron in gray. The region in light gray is not observed in this study. The anticodon-intron (A-I) base pair is in yellow. The Watson-Crick base pairs are indicated with lines, and the G-U wobble is indicated by a dot. The A-I stem, the 3' and 5' splice sites are labeled. Elbow region is formed by the interactions between the D-loop and T-loop. **d–f** Cryo-EM reconstruction at 3.19 Å resolution and an atomic model of human TSEN–pre-tRNA^Arg complex. The pre-RNA is colored in misty rose (**d**, **f**) or as in panel (**c**) (**e**). The Cryo-EM map in panel (**f**) is viewed after a 90° rotation around the horizontal axis. The position of Lα is labeled. Structure figures are produced with Chimera[42] and ChimeraX[47].

cleavage[24–26]. Mutations in the human TSEN complex and its regulatory factor Clp1 are linked to a range of neurodegenerative diseases, including various forms of pontocerebellar hypoplasia (PCH) and intellectual disability[27,28].

Nonetheless, the molecular details of how the functional human tRNA splicing endonuclease complex assembles, recognizes pre-tRNA, and precisely carries out cleaving remain unclear due to the lack of structural information. In this work, we report the structure of an active human tRNA splicing endonuclease complex bound to an intron-containing pre-tRNA, revealing the recognition and cleavage mechanism of pre-tRNA by human TSEN endonuclease complex.

## Results

### Structure determination and overall structure of the machinery

To investigate the mechanisms involved in pre-tRNA intron removal, we prepared a fully recombinant TSEN endonuclease complex by co-expressing wild-type full-length human TSEN54, TSEN34, TSEN2, and TSEN15 in baculovirus-infected insect cells. We observed efficient cleavage activity by mixing it with human pre-tRNA^Arg (anticodon TCT-2-1), which contains a 15-nt intron located one nucleotide to the 3' end of the anticodon triplet (Fig. 1b, c and Supplementary Fig. 1a). We further

confirmed the formation of an active TSEN–pre-tRNA^Arg complex by gel filtration chromatography and analyzed the sample states by SDS-PAGE and denaturing TBE-Urea gels (Supplementary Fig. 1b). We observed clear tRNA cleavage products. To obtain more homogeneous samples and capture the in-action structure of the wild-type complex, we reduced the incubation time, omitted the subsequent gel filtration step, and kept the sample at 4 °C or on ice, which preserved the integrity of most pre-tRNA molecules during the process of freezing the grid (Supplementary Fig. 1a). This way helped minimize the cleavage reaction and allowed us to observe the pre-tRNA trapped in the endonuclease active site. After processing approximately 5 million particles, we were able to obtain a cryo-EM reconstruction at 3.19 Å resolution with 6% particles (Fig. 1d–f, Supplementary Fig. 2 and Supplementary Table 1).

The overall structural arrangement of the human TSEN complex is in a rectangular parallelepiped pattern, similar to that of the α'₂-type endonuclease from *Archaeglobus fulgidus* bound with a BHB RNA (Fig. 1e and Supplementary Fig. 3)[20]. The pre-tRNA is positioned on one side of the endonuclease structure. Several surface segments of the TSEN subunits and the intron loop sequence were not included in the atomic model due to a lack of density, suggesting that these regions form flexible modules (Fig. 1a, c). TSEN54 and TSEN15 function as

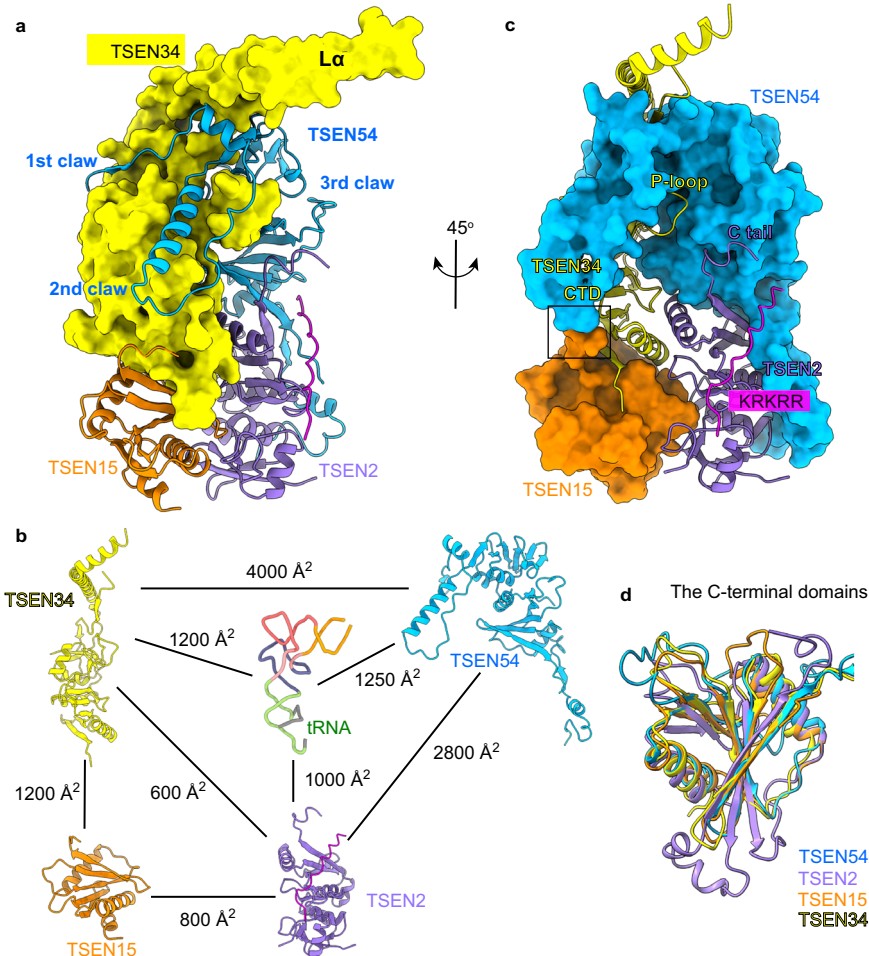

**Fig. 2 | Assembly mechanism of human TSEN complex. a** Interfaces between human TSEN34 with the N-terminal extension of TSEN54. TSEN34 is shown as molecular surface. The three claws of TSEN54 are labeled. **b** Buried surface areas at the interfaces in human TSEN–pre-tRNA complex. **c** Interfaces between the subunits of human TSEN. Molecular surfaces of TSEN54 and TSEN15, colored in blue and dark orange, respectively. The minimal interface between TSEN54 and TSEN15 is indicated in the black box. The positions of P-loop of TSEN34, the C tail, and the KRKRR motif of TSEN2 are labeled. The N-terminal insertion of TSEN2 is in magenta. **d** Overlay of the C-terminal domains of TSEN subunits.

scaffolds to recruit the catalytic subunits TSEN34 and TSEN2. TSEN54 makes extensive contacts with TSEN34 in a 'three-claw-clamp' conformation, tightly holding the body of TSEN34 with ~4000 Å² buried surface area (Fig. 2a, b). Particularly, the N-terminal extension (NTE, residues 1–75) of TSEN54, which forms the first two claws, is well-conserved in eukaryotes, and contributes more than half (~2500 Å²) of the buried surface area. TSEN54 also interacts with its neighboring TSEN2, as well as TSEN34 with its neighboring TSEN15, through their C-terminal β-strands (Figs. 1e and 2a). These interactions bury ~2800 Å² and ~1200 Å² of surface area in the interface, respectively (Fig. 2b). These findings are in agreement with the recent X-ray crystallographic structure of the TSEN34-TSEN15 heterodimer[14]. The interface between two structural subunits, TSEN54 and TSEN15, is minimal (Fig. 2c). The C-terminal domains (CTD) of TSEN34 and TSEN2 contain the catalytic region and are arranged close to each other (Fig. 2c), which is the structural basis of the conserved recognition mechanism known as the 'cross-subunit cation-π-sandwich' pattern. Remarkably, the CTDs of these four subunits have a similar overall structure, all folded into an α/β structure reminiscent of the Rossman fold (Fig. 2d). Their domain architecture and arrangement are similar to those of the archaeal EndAs[12,20,29], suggesting a close evolutionary relationship between archaeal and eukaryotic endonucleases.

The three-dimensional structure of pre-tRNA is in an L-shape conformation, similar to that of mature tRNA (Fig. 3a and

Supplementary Fig. 4a). The intronic sequence near the 3′ splice site produces the anticodon-intron (A-I) base pair with nucleotides in the anticodon loop of mature tRNA, starting with C32:G50 (Fig. 3b, Supplementary Figs. 4b and 5a). These base pairs fold into an A-form 4-base pair helix that is similar to the structure of the BHB in archaea[20]. Interestingly, the A-I stem stacks on the anticodon stem to form a coaxial helix, causing three consecutive bases to be flipped out of the RNA helix (Supplementary Fig. 4b, c). The three flipped-out bases, C51, A52, and A53, create a bulge, called the 3′ splice-site bulge, which flanks the A-I helix (Supplementary Figs. 4c and 5a). Based on the cryo-EM density map and RNA sequence analysis, we found that the nucleotides following the opposite end of the A-I stem form a single-strand loop with high flexibility (Fig. 1c and Supplementary Fig. 4), which explains why the TSEN complex can function with varying intron lengths. Furthermore, we hypothesize that the nucleotides U36, A37, and G38 around the 5′ splice site may be arranged similarly to the 3′ bulge, although only the density of G38 is clearly visible in our study (Fig. 1c and Supplementary Fig. 5). We will discuss this further later.

### Recognition of the mature body of pre-tRNA
The recognition of eukaryotic pre-tRNA includes two parts: the mature body of pre-tRNA and the anticodon-intron region with splicing boundaries, which contain the 3′ splice-site bulge and the 5′ splice site. According to detailed structural analysis and the surface charge

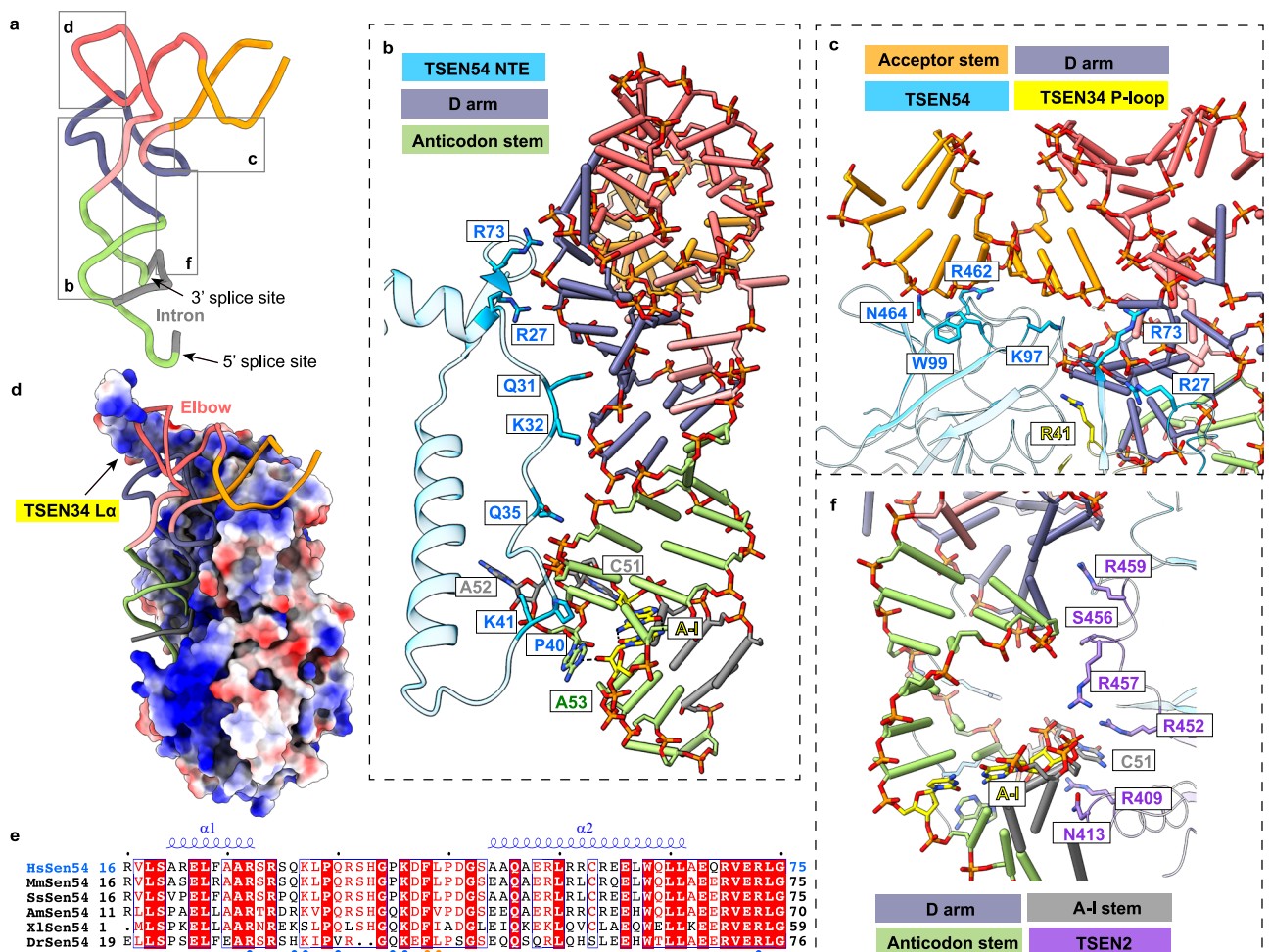

**Fig. 3 | Recognition of the mature body of pre-tRNA. a** The 3D structure of pre-tRNA^Arg^. The color scheme is the same as in Fig. 1c. Close-up views depicting the interactions between pre-tRNA and proteins are shown in panels (**b–d**) and (**f**). **b** Detail view of D arm and anticodon stem recognition by the NTE of TSEN54. Regions (residues 20–75) of TSEN54 NTE are displayed. Key residues that are involved in recognition are shown as sticks. The nucleobases in the A-I base pair (yellow) and the 3' bugle (green and gray) are shown as sticks. The last bulge nucleotide, A53, forms hydrogen-bonding interaction with the A-I base pair. **c** Recognition of the acceptor stem by TSEN54. R41 from P-loop of TSEN34 is also

indicated. **d** Electrostatic surface of the TSEN complex, showing charged interface with the pre-tRNA. The positions of the elbow region and TSEN34 Lα are labeled. **e** Sequence alignment of the N-terminal insertion of selected vertebrate TSEN54 homologs. The secondary structure elements in the human TSEN54 structure are shown. Strictly conserved residues are marked in white with a red background, and well-conserved residues in red. The region that interacts with pre-tRNA is indicated in blue dots, and that with TSEN15 in orange dots. Hs: human; Mm: mouse; Ss: pig; Am: alligator; Xl: frog; Dr: zebrafish. **f** Interfaces between TSEN2 and pre-tRNA.

distribution, we discovered that TSEN54, TSEN34, and TSEN2 work cooperatively to recognize the mature body shape of pre-tRNA with an intricate network of interactions in a sequence-independent pattern (Fig. 3 and Supplementary Fig. 6). TSEN54 contributes the largest interaction surface with the RNA substrate, ~1250 Å² to the buried surface area (Fig. 2b), while the unique NTE contributes 650 Å². The residues 27–48 in the NTE form an extended loop parallel to the coaxial helix of the D-anticodon-intron stem, forming several hydrogen-bonding contacts with the D arm and the anticodon stem (Fig. 3b and Supplementary Fig. 6b). The N-terminal domain (NTD) and CTD of TSEN54 bind the acceptor stem of the pre-tRNA (Fig. 3c and Supplementary Fig. 6c). For TSEN34, it's responsible for sensing three distinct regions of the tRNA molecule: the D arm, the elbow region, and the 3' bulge. Within these regions, a protruding loop (P-loop, residues 33–48) touches the D arm via R41 (Fig. 3c and Supplementary Fig. 6c). The elbow region, which is one main feature of the mature tRNA body, is surprisingly surrounded by a long α helix (labeled Lα, residues 78–128) of TSEN34. Although the density for this helix is relatively weak, the first half of Lα sits on TSEN54, while the second half (residues 105–128) directly senses the D-loop and T-loop of tRNA through ionic

and hydrophilic interactions in nature (Figs. 2c and 3d). The NTE of TSEN54 and Lα of TSEN34 are two major structural differences between the archaeal and eukaryotic tRNA splicing complexes, with high sequence conservation and similar AlphaFold models observed in eukaryotes (Fig. 3e and Supplementary Fig. 7), indicating that these two specific structural elements are crucial for recognizing and locking the mature body of pre-tRNA in place. TSEN2 also contributes to sensing the D stem by its C tail, S456 and R459, and clamping the intron in the A-I stem through N413 and R457 (Fig. 3f and Supplementary Fig. 6d). TSEN15 has no direct interactions with the RNA substrate in the model we present here (Fig. 2b).

## Recognition and cleavage mechanism of the 3' splicing boundary

The two splicing boundaries are positioned in clearly defined pockets formed by the interface of two catalytic subunits, TSEN34 and TSEN2 (Fig. 4a). In the case of the 3' splice-site bulge, the three bases are well-ordered and point in three different directions, generating a sharply bent backbone (Fig. 4b). The cation-π interactions with TSEN2 R409 and R452 sandwich the first bulge base, C51. The nitrogen atom of the

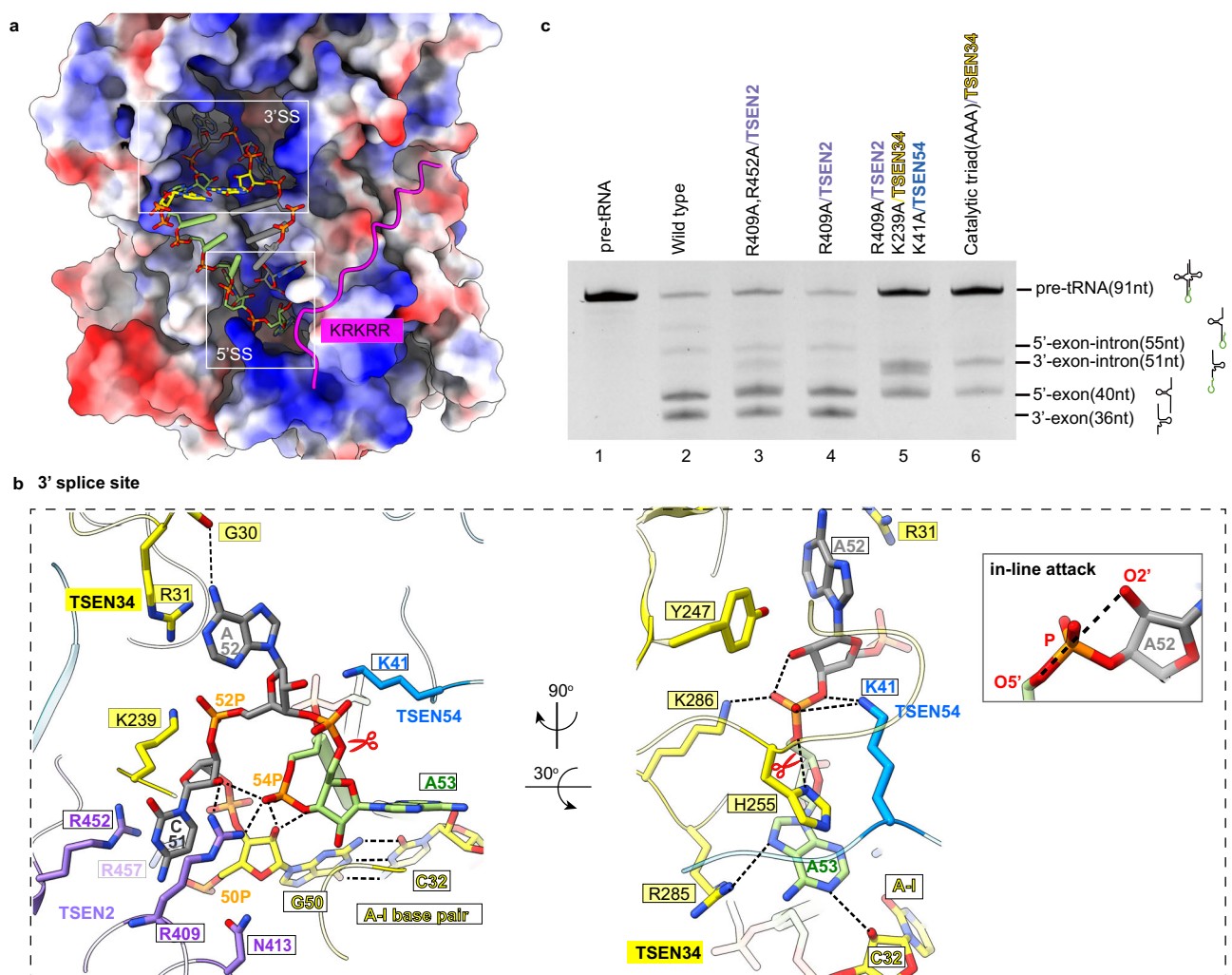

**Fig. 4 | Recognition and cleavage mechanisms of 3′ splicing boundary by human TSEN complex. a** Binding pockets for 3′ and 5′ splicing boundaries of tRNA molecules. Electrostatic surface of the body of the TSEN complex shows charged pockets. The A-I base pair is shown as yellow sticks. The nucleotides of the anticodon loop are in green, and those of the intron are in gray. The two splicing boundaries, shown as sticks, are located in deep pockets in TSEN. The N-terminal insertion of TSEN2, shown as cartoon in magenta, covers part of the 5′ splice center. The position of the positively charged motif is labeled. 3′SS: 3′ splice site; 5′SS: 5′ splice site. **b** Detail views of the 3′ splice site with TSEN34, TSEN2, and TSEN54. The 3′ bugle and A-I base pair are shown as sticks. C51 is cation-π interactions with TSEN2 R409 and R452. R409 forms hydrogen bonds with the backbones of both C51 and U54. TSEN2 R457 and N413 and TSEN34 R31 and K239 interact with the phosphate groups of G50 and A52, respectively. In the right most panel, the A53

base is surrounded by H255, R285, and the A-I base pair from three sides. The scissile phosphate, indicated with the symbol 'scissors', is located in the center of the catalytic triad of TSEN34 –Y247, H255, and K286, covered by TSEN54 K41 at the same time. The insert presents the in-line geometry created by the attacking 2′-nucleophile oxygen, scissile phosphate, and the leaving 5′-oxygen. **c** Cleavage assays showing that the R409A mutation alone had minimal impact (lane 4), whereas the double mutants R409A and R452A, resulting in the accumulation of the 3′-exon-intron intermediate, decreased the cleavage rate (lane 3). The triple mutants R409A (TSEN2), K239A (TSEN34), and K41A (TSEN54) completely abolished cleavage (lane 5), similar to mutating the catalytic triad of residues to Ala in TSEN34 (lane 6). The cleavage assays were repeated three times. Source data are provided as a Source data file.

second base, A52, is recognized by a hydrogen bond with the main-chain carbonyl of G30 in TSEN34. The third nucleotide, A53, is π-stacked with the side chain of H255 on one face and stabilized on the other two sides by the minor groove of the A-I stem and R285 of TSEN34 (Fig. 4b, right panel). Although similar recognition mechanisms have been observed in archaeal EndA (Supplementary Fig. 8)[20], the human TSEN-RNA complex has a more complex interaction network. A series of basic residues, such as R31 and K239 of TSEN34, contribute to recognizing the ribose-phosphate backbone of the bulge (Fig. 4b). Basic residue R31 is absent in the α′₂-type *A. fulgidus* EndA at this position, while a conserved lysine was observed near this site in the (αβ)₂-type *Aeropyrum pernix* EndA (Crenarchaea specific loop, CSL) and the ε₂-type ARMAN-2 EndA (ARMAN-2 specific loop, ASL)[12,29,30]. The (αβ)₂ and ε₂ EndAs both show broad substrate specificity, in contrast to the narrow substrate specificity of the α′₂ type, which

indicates that the TSEN complex may also have broad substrate specificity. The conserved R409 of TSEN2 forms hydrogen bonds with the backbones of both C51 and U54 (Fig. 4b), being the important structural basis for the formation of a sharply bent bulge for cleavage. In addition, the unique NTE of eukaryotic TSEN54 folds over the 3′ splicing boundary and touches the scissile phosphate through K41 (Fig. 4b and Supplementary Fig. 9), further trapping the bugle base in the catalytic center of the 3′ splice site and supporting efficient cleavage.

The conformation of the endonuclease-RNA complex observed here is poised for the cleavage reaction. The catalytic triad of residues in TSEN34, Y247, H255, and K286 adopt similar binding modes with the scissile phosphate compared to the structure in archaea (Supplementary Fig. 8)[20,31]. The in-line geometry of the attacking 2′-oxygen, the scissile phosphate, and the leaving 5′-oxygen is formed (Fig. 4b, insert). The side chain of Y247 induces deprotonation of the 2′-OH group,

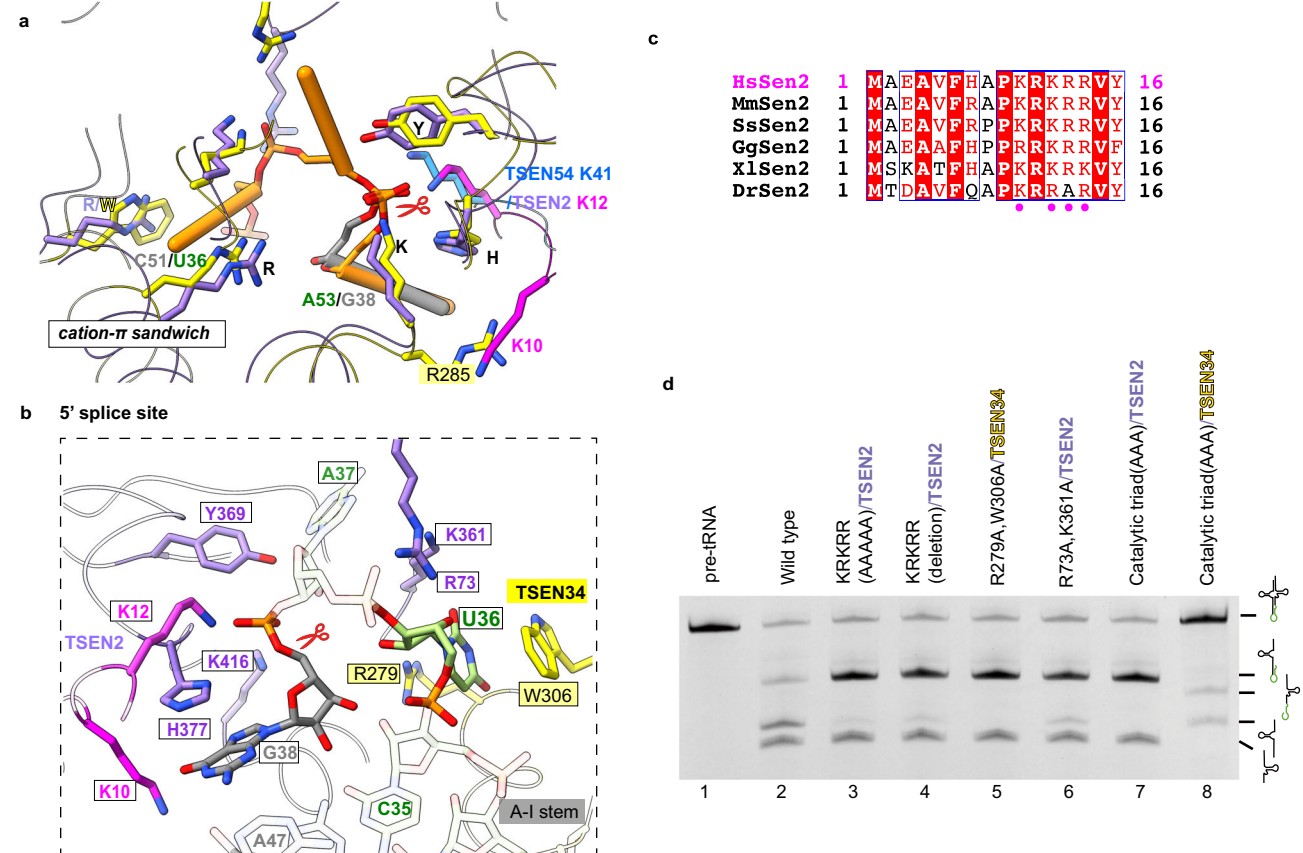

**Fig. 5 | Recognition and cleavage mechanisms of 5′ splicing boundary by human TSEN complex. a** Overlay of the catalytic centers of TSEN34 and TSEN2. The 3′ bugle in TSEN34 is shown in orange, while the nucleotide G38 in TSEN2 is in gray. TSEN2 was used for the overlay. Key residues that are involved in recognition and cleavage are shown. K10 and K12 in the KRKRR motif of TSEN2 play similar roles for the 5′ splice site as R285 of TSEN34 and K41 of TSEN54 for the 3′ splice site. **b** Detail view of the 5′ splice site with TSEN34 and TSEN2. U36 interacts with TSEN34 R279 and W306 via cation-π interactions. The scissile phosphate is located in the center of the catalytic triad of TSEN2 –Y369, H377, and K416. A37 that is not visible in the density map is shown as transparent sticks. **c** Sequence alignment of the N-terminal insertion of selected vertebrate TSEN2 homologs. Well-conserved residues are highlighted in red. The positively charged motif, close to the intron loop, is indicated in magenta dots. Hs: human; Mm: mouse; Ss: pig; Gg: chicken; Xl: frog; Dr: zebrafish. **d** Cleavage assays were carried out to assess the impact of various mutations on the cleavage activity at the 5′ splice site. The mutation of the catalytic triad of residues in TSEN2 (lane 7) resulted in the complete loss of cleavage activity. Deleting the KRKRR motif of TSEN2 or mutating it to AAAAA greatly reduced the cleavage activity at the 5′ splice site (lanes 3 and 4). The mutation of the R279-W306 tweezer in TSEN34 led to the complete abolishment of cleavage at the 5′ splice site (lane 5). Mutating R73A and K361A in TSEN2 slowed down the cleavage rate (lane 6). For reference, lane 8 represents the result obtained from mutating the catalytic triad of residues in TSEN34. These cleavage assays were repeated three times. The reactions were carried out at 37 °C for 10 min and the molar ratio of RNA to TESN was 1:1. Source data are provided as a Source data file.

initiating the nucleophilic attack on the adjacent phosphorus. The leaving group, 5′ oxygen, is protonated by H255. K286 is responsible for stabilizing the transition state (Fig. 4b)[20]. Our structure indicates that archaeal and eukaryotic endonucleases share a conserved catalytic mechanism for the removal of the tRNA intron.

Our structural observations are supported by the mutational analysis (Fig. 4c and Supplementary Fig. 10). The cleavage activity for the 3′ splice site was completely abolished by mutating the catalytic triad of residues to Ala in TSEN34 (lane 6). While the R409A mutation alone had minimal impact (lane 4), the double mutants R409A and R452A, which affect the cation-π recognition of the first bulge base, exhibited a slower cleavage rate, leading to the accumulation of the 3′-exon-intron intermediate (lane 3). Interestingly, the residues R409A (TSEN2), K239A (TSEN34), and K41A (TSEN54) are involved in capturing the ribose-phosphate backbone in the catalytic pocket. We found that introducing the K41A mutation in TSEN54 or the R409A mutation in TSEN2 enhances the effect of the double mutants formed by R409$^{TSEN2}$/K239$^{TSEN34}$ or K41$^{TSEN54}$/K239$^{TSEN34}$, resulting in a complete loss of cleavage. This observation strongly suggests that the residues R409$^{TSEN2}$, K239$^{TSEN34}$, and K41$^{TSEN54}$ play crucial roles in stabilizing the

conformation of the 3′ bulge (Fig. 4c, lane 5 and Supplementary Fig. 11a).

## Recognition and cleavage mechanism of the 5′ splicing boundary

At the 5′ splicing boundary, we observed two nucleotide densities stacking with H377 and W306, respectively (Supplementary Fig. 5e–g). Due to the disrupted densities with neighboring nucleotides, we are unable to definitively assign these two nucleotides based solely on the densities. However, when overlaying TSEN34 and TSEN2, which share a similar architecture, we observe an overlap of positions for both the key residues in the enzymes and the RNA, as well as the cover loops, K41 in TSEN54 and KRKRR in TSEN2 (Fig. 5a). Thus, the interaction network at the 5′ splice site in our structure exhibits high similarity to that of the 3′ splice site and to that found in the archaeal RNA-splicing endonucleases. Given that G38 is the cleavage site, it is likely that the nucleotide stacking with H377 corresponds to G38, which also brings the one stacking with W306 assigned to U36. The strong density of the scissile phosphate indicates that the 5′ splice boundary is also ready for cleavage (Supplementary Fig. 5f, g). The scissile phosphate is located at

the center of the catalytic triad of residues in TSEN2, Y369, H377, and K416 (Fig. 5b and Supplementary Fig. 5f, g). Similarly, G38 is stabilized through π-stacked with H377 and by the A-I stem and the KRKRR motif of TSEN2. The positively charged KRKRR motif in the N-terminal insertion of TSEN2, which is common in most eukaryotic TSEN2 but lacking in archaea, covers up the active site of TSEN2 (Figs. 4a, 5c and Supplementary Fig. 5h). K10 and K12 in the KRKRR motif play similar roles for the 5′ splice site as R285 in TSEN34 and K41 in TSEN54 for the 3′ splice site (Fig. 5a). Our structure reveals that the KRKRR motif of TSEN2 helps stabilize the nucleotides surrounding the 5′ splice site into the TSEN2 catalytic pocket. However, A37, corresponding to A52, is missing due to the flexibility, while U36 is recognized by W306 and R279 in TSEN34 (Fig. 5b and Supplementary Fig. 5e). Previous research in the yeast Sen complex has shown the critical roles of the Trp-Arg tweezer pair in recognizing and catalyzing the 5′ splice site[22]. The weak density of U36 in our map (Supplementary Fig. 5e) is possibly due to the fact that the 5′ splicing boundary is solvent-accessible, unlike the deeply buried 3′ bulge (Fig. 4a and Supplementary Fig. 9). This accessibility may cause the conformation of the 5′ splicing boundary to be less stable, resulting in a transient interaction between U36 and the R279-W306 tweezer.

To verify our structural findings, we carried out mutagenesis and biochemical experiments (Fig. 5d and Supplementary Fig. 10). In comparison to the wild-type TSEN, the mutation of the catalytic triad of residues in TSEN2 abolished the cleavage activity at the 5′ splice site (lane 7). Notably, deleting the KRKRR motif of TSEN2 or mutating it to AAAAA greatly reduced the cleavage activity at the 5′ splice site, resulting in the accumulation of the 5′-exon-intron intermediate and fewer 5′ exon products (lanes 3 and 4). This observation supports the important role of the KRKRR motif in stabilizing the negatively charged intron. Remarkably, despite the poor density of U36 and A37, the mutation of the R279-W306 tweezer in TSEN34 led to the complete abolishment of cleavage at the 5′ splice site (lane 5; Supplementary Fig. 11b), which differs from the mild impact observed with the mutation of the R409-R452 tweezer at the 3′ splice site (Fig. 4c, lane 3). Furthermore, mutating R73A and K361A residues in TSEN2 significantly slowed down the cleavage rate, indicating their potential involvement in transiently stabilizing the phosphate backbone of A37 during cleavage (lane 6). Hence, our biochemical studies provide support for the structural observations. Similar to the 3′ splice site, the residues for the recognition of the first bulge base (R279-W306 tweezer), the stabilization of the ribose-phosphate backbone (R73 and K361), and the cleavage process (the catalytic triad) are important for the 5′ splice site. In addition, the KRKRR motif in the N-terminal insertion of TSEN2 is unexpectedly required for highly efficient cleavage. These findings from the mutational analysis enhance the overall understanding of the mechanisms involved in the cleavage process of the 5′ splice site.

## Discussion

The positions of the A-I base pair and 3′ splice-site bulge are strictly conserved in human pre-tRNAs (Supplementary Fig. 12), consistent with the limited space of the binding pocket for the 3′ splicing boundary. Therefore, the recognition and cleavage mechanisms elucidated in this study can also be applied to other intron-containing pre-tRNAs in humans. The 'cross-subunit cation-π sandwich' (R409 and R452 in TSEN2, but R279 and W306 in TSEN34) and the A-I stem are fundamental to both 3′ and 5′ splice reactions of intron-containing tRNA in eukaryotes, which are evolutionarily conserved with those in archaea[20]. Interestingly, our findings appear contradictory to previous studies on the yeast Sen complex, where it was concluded that the cleavage of the 3′ splice site is independent of the cation-π recognition by Sen2[22]. However, through sequence alignment and utilizing the AlphaFold model of yeast Sen2, we identified R321 and R369 of yeast Sen2 (instead of R321 and W348 in earlier biochemical studies[22]) at

positions corresponding to the arginine tweezer pair (R409 and R452) in human TSEN2. This finding suggests that the arginine tweezer pair at the 3′ splice site is conserved in the yeast Sen complex as well. In fact, our research, as well as others, has confirmed that eukaryotic splicing endonucleases can recognize and splice a pre-tRNA mimic that only contains the anticodon domain with a Bulge-Helix-Loop motif (Supplementary Fig. 11c)[32,33]. In the eukaryotic tRNA splicing complex, the additional specialized structural characteristics, such as the NTE of TSEN54, the Lα of TSEN34, and the C tail of TSEN2, cooperatively recognize specific regions of the mature body of pre-tRNA. This sheds light on why eukaryotic TSEN requires the presence of the mature domain of pre-tRNA for efficient cleavage, ultimately enhancing its ability to effectively select RNA substrates in vivo. The K41 in TSEN54 and the KRKRR motif in the N-terminal insertion of TSEN2 support efficient cleavage (Supplementary Fig. 9). However, it is noteworthy that, based on our structural model, none of the PCH-associated mutations identified to date reside within the pre-tRNA binding interfaces. This implies that these mutations are more likely to impact the structural stability or subunit folding of TSEN and thereby contribute to disease development. This observation is consistent with previous results from differential scanning fluorimetry analyses, which showed that PCH-associated mutations cause thermal destabilization[14].

Another cryo-EM structure of the human TSEN bound to pre-tRNA has recently been published[34], as well as other publications of the same complex (PDB ID: 7UXA and PDB ID: 7ZRZ)[35,36]. These studies have provided similar insights into the recognition mechanism of the mature body of pre-tRNA and the molecular process of 3′ splice site cleavage. Notably, the conformations of the 5′ splice site in different structures varied. In the case of mutating all three catalytic residues to Ala in TSEN2, no RNA density was observed in the binding pocket of the 5′ splice site[35] (Supplementary Fig. 13a, b). The crucial residue K416 exhibited a different conformation, being distant from the active site when using truncated TSEN with the H377F mutation (Supplementary Fig. 13c). Similarly, in the TSEN complex with TSEN2 H377A mutant, the distance between the active site and the scissile phosphate of G38 is too far (Supplementary Fig. 13d). For the structure of wild-type TSEN in the post-catalytic state, the sample revealed almost complete cleavage of the products, resulting in the absence of density for the scissile phosphate (Supplementary Fig. 13e)[34]. Interestingly, the complex structure presented in this study differed from both the catalytic-dead state and the post-catalytic state. This is supported by the presence of G38 and the NTE loop of TSEN2 (Supplementary Fig. 5f–h), a critical feature missing in all other structural studies. It is speculated that the NTE of TSEN2 is flexible in structure until it cooperatively traps G38 with the catalytic triad of residues in TSEN2, thereby locking the scissile phosphate into the catalytic pocket. All these studies collectively provide compelling evidence that any catalytic mutation in TSEN2 not only abolishes the cleavage activity but also weakens the binding affinity between TSEN2 and the 5′ splice site of pre-tRNA, which is distinct from the binding pocket of the 3′ splice site. Interestingly, while the catalytic mutation in TSEN34 also abolishes the cleavage activity, it does not impact the conformation of the 3′ bulge. Additionally, our mutagenesis and biochemical experiments have provided further insights into the functional roles of specific residues in TSEN. We found that double mutants of the R279-W306 tweezer completely abolished cleavage at the 5′ splice site (Fig. 5d), whereas mutations of the R409-R452 tweezer retained activity for the 3′ splice site (Fig. 4c), indicating that the correct conformation of the 5′ splice site is strictly dependent on the combined contributions of both the R279-W306 tweezer and the catalytic triad residues of TSEN2. Following studies are needed to further confirm the sequence assignment at the 5′ splice site with optimized sample preparation and image processing protocols.

Overall, our structural and biochemical studies have shed light on the molecular mechanisms underlying pre-tRNA recognition and

processing by wild-type TSEN machinery in humans. TSEN54 and TSEN15 function as scaffolds, bringing together the catalytic subunits TSEN34 and TSEN2 to form a compact structure. TSEN54 acts as a molecular ruler, along with unique regions of TSEN34 and TSEN2, cooperatively interacts with the intron-containing pre-tRNA to lock it in place and guide the A-I stem to the correct position. The catalytic reaction at 3′ and 5′ splice sites is carried out by a conserved mechanism in archaeal and eukaryotic endonucleases. While many insertions in the TSEN machinery are not involved in our model due to the flexibility. These regions may recruit other regulatory factors for metazoan tRNA processing, Clp1 for instance[34,36]. Moreover, our findings also contribute to understanding the pathogenic mechanism of PCH. Further studies are needed to investigate how the non-canonical introns, particularly the extremely short ones, are spliced at the molecular level.

## Methods

### Protein expression and purification

The full-length human TSEN complex was expressed in insect cells using Multibac technology. TSEN15 and TSEN2 were cloned into the pFL acceptor vector, and a 6 x His tag was added to the N terminus of TSEN15. TSEN34 was cloned into the pSPL donor vector. The donor was fused to the acceptor by Cre recombinase. N-terminal GST-tagged human TSEN54 was cloned into the pFL vector. Bacmids for all the complexes were generated in DH10Bac(YFP) competent cells by transformation. Baculoviruses were generated by transfecting bacmids into Sf9 cells using FuGENE (Promega). P1 viruses were cultured at 27 °C for 4 days, and P2 viruses for large-scale infection were amplified from P1 viruses in 50 mL Sf9 cells at 27 °C for 3 days. One liter of High5 cells ($1.8 \times 10^6$ cells ml$^{-1}$) cultured in ESF 921 medium (Expression Systems) was infected with a suitable amount of P2 virus at 27 °C with constant shaking. Cells were harvested after 48 h by centrifugation at $2000 \times g$ for 15 min. More cloning information can be found in Supplementary Data 1.

For purification, the cell pellet was re-suspended in 100 mL buffer containing 50 mM HEPES (pH 8.0), 350 mM NaCl, 10% glycerol, 10 mM imidazole, and 1 mL protease inhibitor cocktail (EDTA-Free, 100X in DMSO; ApexBio), followed by sonication to lyse the cells. The cell lysate was then centrifuged at $24,300 \times g$ for 60 min at 4 °C. The supernatant was incubated with nickel beads for 1 hour at 4 °C. The beads were then washed 4 times with 50 bed volumes of wash buffer (20 mM HEPES (pH 8.0), 350 mM NaCl, 10% glycerol and 30 mM imidazole) and eluted with a buffer containing 20 mM HEPES (pH 8.0), 150 mM NaCl, 5% glycerol and 500 mM imidazole. GST tag was removed by TEV protease at room temperature for 6 h. The protein was further purified by chromatography using a HiLoad 16/600 Superdex 200 pg column (Cytivia). The peak fractions were used for EM studies in a buffer containing 20 mM HEPES (pH 8.0), 150 mM NaCl, 1 mM MgCl$_2$, and 5 mM DTT.

### Preparation of RNA substrates

The anticodon stem loop RNA (32 nucleotides, /5-FAM/UUGGACUU-CUAGUGACGAAUAGAGCAAUUCAA) was purchased from GENEWIZ. Full-length pre-tRNA$^{Arg}_{TCT}$ genes were synthesized and cloned into the pUC-GW-kan vector by GENEWIZ as well. DNA templates for transcription were generated using PCR and then extracted by PEG6000. The 5′ primer sequence used was <u>TAATACGACTCACTATAg</u>GGGGC, with the T7 promoter sequence underlined. Pre-tRNAs were amplified in vitro using the T7 High Yield RNA Transcription kit (Vazyme). The site of transcription initiation is indicated in lower case, resulting in the addition of the sequence "GGG" at the 5′-end of pre-tRNA$^{Arg}_{TCT}$, thereby increasing its length to 91 nt. In vitro transcription was carried out with 2 µg DNA templates at 37 °C for 16 h. RNA substrates were separated and concentrated through the HiPure RNA Pure Micro Kits (Magen).

### tRNA intron splicing assay

The RNA substrates were heated at 95 °C for 5 min, followed by a slow cooling to room temperature to allow for annealing. tRNA cleavage reactions were carried out in RNA cleavage buffer (50 mM KCl, 50 mM HEPES pH 8.0, 1 mM MgCl$_2$, 1 mM DTT and 1 unit/mL RNAsin (Promega)). The reactions were quenched by 2x Urea denaturing RNA loading buffer (8 M Urea in 1x TBE, 0.01% (w/v) bromophenol blue). The samples were boiled at 98 °C for 10 min and immediately loaded onto a 15% Urea-TBE gel for 40 min at 250 V. Using pre-tRNA$^{Arg}_{TCT}$ as substrates, the reactions were carried out on ice for 45 min and the molar ratio of RNA to TESN was 1:1. The gels were stained with 4SGelred (Sangon Biotech) for 5 min and imaged on Tanon MINI Space 1000 set to 302 nm. In the case of FAM-labeled anticodon stem loop RNA substrates, the reactions were carried out on ice for 30 min with a molar ratio of RNA to TESN of 1:29. The gels were visualized on Amersham ImageQuant 800 (Cytivia) set to 460 nm.

### Human TSEN−pre-tRNA complex formation

Purified TSEN complex and pre-tRNA$^{Arg}_{TCT}$ were mixed at a molar ratio of 1:2. The reaction mixture was incubated on ice for 1 h and then loaded onto a Superose 6 Increase 10/300 GL column (Cytivia), in a buffer containing 20 mM HEPES (pH 8.0), 150 mM NaCl, 1 mM MgCl$_2$ and 5 mM DTT. The presence of RNA in the complex was confirmed with an A260/A280 ratio of 1.3, while the ratio was 0.5 if RNA was left out of the mixture.

### Cryo-EM sample preparation and data collection

Fractions of interest were directly used for EM studies. The homogeneity of the sample was first examined by negative-stain EM with 0.7% (w/v) uranyl acetate. To prepare grids for cryo-EM, the freshly purified TSEN sample was centrifuged at $13,000 \times g$ for 5 min to remove potential protein aggregates. The protein concentration was measured with a NanoDrop spectrophotometer (Thermo Fisher Scientific) and adjusted to 0.8 mg/mL. After incubating the proteins and annealing pre-tRNAs at a molar ratio of 1:1.1 on ice for 45 min, a 4 µl aliquot was applied to glow-discharged grids (200 mesh R2/1 Au grids, Quantifoil) and frozen with a Vitrobot Mark IV (Thermo Fisher Scientific) set at 4 °C and 100% humidity. After a 5-s incubation, the grids were blotted for 2 s with a blot force of −2 and then plunged into liquid nitrogen-cooled ethane. Data collection was performed on a 300 kV Titan Krios G4 electron microscope (Thermo Fisher Scientific) at a nominal magnification of ×81,000, corresponding to a calibrated pixel size of 1.055 Å at the specimen level. Images were collected using a defocus range of −1.4 to −2.4 µm with a BioContinuum K3 direct electron detector (Gatan) in super-resolution counting mode.

### Image processing

4500 image stacks were motion-corrected and dose-weighted in MotionCor2[37]. The contrast transfer function (CTF) parameters were determined with CTFFIND4[38] implemented in RELION-3[39]. 5,263,873 particles were automatically picked from 4345 micrographs with Gautomatch (https://www.mrc-lmb.cam.ac.uk/kzhang/Gautomatch/) using the selected 2D averages from a small subset of the data as templates. The particles were extracted and normalized into $200 \times 200$ boxes. To avoid the loss of the particles in rare orientations, 2D classification step was skipped and the extracted particles were directly subjected to Relion 3D classification into six classes using the initial model obtained with Cryosparc[40] ab-initial reconstruction. Among the six classes, only one class showed clear structural feature of a complex, while others shows much less secondary structures or the complex is less complete. 1568k particles in the good class were selected and subjected to another round of 3D classification in Cryosparc. One of the four classes contain more densities belongs to TSEN2 (335 K particles, depicted in purple). The additional densities are labeled with a '*' (star). While the rest parts of the complex are

similar with the first two classes. The quality of the fourth class (cyan) is poor. The alignment parameters are optimized in Cryosparc non-uniform refinement to generate a map at 3.44 Å. To further improve the map quality, these 335 K particles were processed with 3D auto-refine, CTF refinement, and Bayesian polishing in RELION-3. After these processes, a map at an overall resolution of 3.19 Å was obtained with Cryosparc non-uniform refinement. To further improve the local resolution of the KRKRR motif, a soft mask near the KRKRR region was applied during the Cryosparc Local Refinement.

## Model building and refinement

We used the predicted structure of human TSEN complex from AlphaFold[41] as the starting model and fitted it into the density map with Chimera[42]. The trace of Lα in TSEN34 is following the AlphaFold model of *Drosophila melanogaster* TSEN34 (AlphaFoldDB Q0E8E7, Supplementary Fig. 7). The structure of tRNA is built based on the combination of the BHB structure in *Archaeoglobus fulgidus* (PDB ID 2GJW) and yeast tRNA$^{Arg}$ (PDB ID 1F7V). All manual model building was performed with Coot[43]. The atomic model was refined by using phenix.real_space_refine[44]. The statistics from the structure determination are summarized in Supplementary Table 1.

## Sequence alignment

Multiple alignment of amino acid sequences was produced with Clustal Omega[45] and rendered with ESPript[46]. Additional annotations were introduced manually.

## Reporting summary

Further information on research design is available in the Nature Portfolio Reporting Summary linked to this article.

## Data availability

The data supporting the findings of this study are available from the corresponding authors upon reasonable request. The atomic coordinates and the EM maps are available in the Protein Data Bank (PDB) at http://www.pdb.org. Accession codes are PDB 8ISS and EMDB-35694. Source data are provided with this paper.

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

## Acknowledgements

We thank the Bio-Electron Microscopy Facility of ShanghaiTech University and the Cryo-Electron Microscopy center at the Interdisciplinary Research Center on Biology and Chemistry, Shanghai Institute of Organic for help with data collection; Xinyi Xiong for initial studies on this project. This research is supported by Shanghai Pujiang Program 21PJ1410600 (to Y.S.), ShanghaiTech University start-up fund 2021F0202-000-02 (to Y.S.), and Shanghai Municipal Science and Technology Major Project 2019SHZDZX02 (to Y.Z.).

## Author contributions

L.Y. carried out protein expression, purification, and biochemical studies. L.Y. and Y.H. carried out EM data collection, analysis, and reconstruction. L.Y. and J.Z. carried out the mutational analysis. Y.S. carried out model building and refinement. Y.Z. and Y.S. supervised the overall project and analyzed the results. Y.Z. and Y.S. wrote the paper, with significant contributions from L.Y. All authors commented on the paper.

## Competing interests

The authors declare no competing interests.
