## [Peer Review File · Nature Communications]

Recognition and cleavage mechanism of intron-containing pre-tRNA by human TSEN endonuclease complexReviewers' Comments:

Reviewer #1:

Remarks to the Author:

Review for Yuan et al.

In their manuscript "Recognition and cleavage mechanism of intron-containing pre-tRNA by human TSEN endonuclease complex", Yuan et al. report the 3.2 Å cryo-EM structure of the human TSEN endonuclease in complex with intron-containing pre-tRNA in a pre-catalytic state.

Overall, this manuscript is well written, well structured, and a pleasure to read. The structural work is competently done and addresses the long-standing question of how human intron-containing tRNAs are specifically recognized and cleaved. It is consistent with previous genetic and biochemical work and supports the claims made by the authors. Thus, I would, in principle, support publication of this article with minor revisions (see minor points below). My only concern is that a recent publication by Zhang et al. in *Molecular Cell* (doi: 10.1016/j.molcel.2023.03.015) already reported the cryo-EM structures of the pre-tRNA-bound human TSEN complex in the pre-catalytic state as well as in the post-catalytic state at resolutions of 2.94 and 2.88 Å, respectively. The pre-catalytic state structure by Zhang et al. and their mechanistic conclusions are very similar to those reported here by Yuan et al., raising the question whether Yuan et al. can provide further insight beyond what has already been published. A central point Yuan et al. might address in more detail in a revision is the fact that the pre-catalytic state structure by Zhang et al. was prepared using enzymatically inactive catalytic site mutants. However, in this context, it would be important to clarify, whether the structure reported here is indeed pre-catalytic and that it does not represent a heterogeneous mix of unprocessed and partially processed pre-tRNA in the post-catalytic state (e.g. present a detail of the electron density around the 3' cleavage site which supports the claim that it is indeed the pre-catalytic state).

Minor points:

- Please provide evidence for the claim that the reported structure represents a homogenous (pre-catalytic) state.
- Line 36: "beautifully" seems a somewhat odd choice of wording in a scientific context.
- Lines 157 and 228: Does the structure provide a clue as to why "additional structural characteristics" in eukaryote TSEN are critical for pre-tRNA recognition, whereas they are dispensable in Archaea?

Reviewer #2:

Remarks to the Author:

In this manuscript, Yuan et al. present the cryo-EM structure of human tRNA splicing endonuclease (TSEN) complexed with an intron-containing pre-tRNA. The structure elucidates the organization of the four subunits within the intact assembly and their role in recognizing and cleaving the pre-tRNA. Notably, the presence of insertion loops originating from TSEN54 and TSEN2, which coordinate the splicing sites, adds an intriguing aspect. Additionally, a comparative analysis between human TSEN and its archaeal counterparts highlights the conserved structural features shared across kingdoms. The structural data and insights presented here overlap significantly with a recent work (Zhang et al. 2023, *Molecular Cell*), but contain unique findings. Overall, this study is of high quality and provides valuable insights into pre-tRNA splicing. However, several points need to be addressed before publication in *Nature Communications*.

Major:

-page4, line 94: The authors state that a cryo-EM reconstruction was obtained using only 6% of the input particles. This raises the question of whether the samples are heterogeneous. However, the details of the data processing pipeline (fig. S2) are not clearly described. Specifically, the heterogeneous refinement resulted in four 3D classes that appear highly similar. Are these classes representing distinct working states of the complex? It is recommended that the authors provide

additional information regarding the data processing steps and elaborate on the criteria used for selecting particles for the reconstruction.

-page6, line132-134: The authors propose that the nucleotides near the 5'SS may form a similar arrangement with 3'SS bulge. I not convinced by this analysis since the cryo-EM map quality in this region is rather poor (fig. S4). The author should provide clear EM density maps to support their structural findings. Indeed, there's only one panel in fig. S4 that shows the EM map, which is insufficient. The authors should provide more details of the EM maps for the RNA conformation as well as the critical interactions described in their main text.

-page6, line 136-141: The authors discussed the recognition of the mature body of pre-tRNA in this paragraph. However, the figures presented here are of low quality, making it challenging to identify the specific interactions mentioned in the text. To support their findings, it is crucial for the authors to provide the EM maps that justify their conclusions. Furthermore, model validations, model-to-map FSC curves, should also be included during revision.

-page8, line 194-216: This paragraph describes the conformation of 5'SS and the interactions that contributes to its recognition. However, the poor EM density maps shown in fig. S4 does not support the modeling of 5'SS. Indeed, there's almost no discernible EM density for U36-G38. We are also not shown by any EM density maps for the critical residues from TSEN2, including the "KRKRR" motif, which are implicated in 5'SS recognition. To justify their model, the author may improve the local resolution of this region and provide the EM density maps in the supplementary information. Otherwise, the description on 5'SS is purely speculative and should be removed from the text.

Minor:

-page4, line71: Reference is required here.

-page6, line 128: "Cryo-EM" should be "cryo-EM"

Reviewer #3:

Remarks to the Author:

Yuan et al. report RNA recognition and cleavage mechanism during tRNA splicing mediated by human TSEN endonuclease based on a single cryoEM structure. Though this work is less thorough than that recently published by Zhang et al. (<https://doi.org/10.1016/j.molcel.2023.03.015>) or other publications in press of the same complex (PDB ID: 7uxa for instance), it provides some unique information lacking in these other structures. However, it should address the following concerns.

1) It is possible that authors did not realize the publication by Zhang et al. (PDB id: 8hzmz and 8hmy) or the structure 7uxa. However, since all these structures are nearly identical with slightly different constructs and catalytic mutations or with CLP1, authors should provide a comparison and point out the differences and similarities in both writings and as supplementary figures. For instance, 8hzmz is also active form but its 3' splice site is cleaved and modeled with the product. But in this work, the 3' splice site is intact. Can authors address why this is the case? Interestingly, authors captured the N terminal extension of TSEN2 near the 5' splice site while other structures did not, thereby providing some unique insights. Thus the comparison should help to emphasize the common conclusions while highlight unique contributions.

2) Various structures captured partial 5' splice site and none seems to show correct positioning of the scissile phosphate in the predicted active site. This is also a good region where authors can compare and comment. Did their structure show correct placement of 5' scissile phosphate and if so, do active site residues cooperate with the location? If not, what is the likely reason?

3) The work is mostly structural with almost no functional verification. Given that some of the findings are unique, it would be useful to verify structural findings by mutagenesis. For instance, the lysine residues one TSEN54 near 3' splice site (K41) and that of TSEN2 (K12, or KRKRR) seem to play quite

an important role in possibly stabilizing the negatively charged intermediate. Mutational analysis would be very helpful.

4) In the intact 3' splice site that is not cleaved, do authors see the proposed inline geometry? If so, please point it out and mark it so.

5) Line 55, please revise the statement "In archaeal EndA, the complex is composed of four subunits that form different architectures, including α_4 , α'_2 , $(\alpha\beta)_2$, or ϵ_2 -type complexes(11, 12)," In fact, the endonucleases have the same architecture that is arranged by using different domains and subunit composition. Some do not have four subunits.

6) Please specify which restriction enzyme was used to linearize the pre-tRNA-encoding plasmid for transcription;

7) The rotamer outlier content is high for this resolution, so is core clash and Ramachandran outliers. Please fix them.

8) Report min/mean/max B factors and model FSC resolutions

9) Please label the axes of the FSC plot;

10) Please label the unit for the local resolute plot color bar

Responses to reviewers' comments (our responses are in black)

Reviewer 1:

In their manuscript “Recognition and cleavage mechanism of intron-containing pre-tRNA by human TSEN endonuclease complex”, Yuan et al. report the 3.2 Å cryo-EM structure of the human TSEN endonuclease in complex with intron-containing pre-tRNA in a pre-catalytic state.

Overall, this manuscript is well written, well structured, and a pleasure to read. The structural work is competently done and addresses the long-standing question of how human intron-containing tRNAs are specifically recognized and cleaved. It is consistent with previous genetic and biochemical work and supports the claims made by the authors. Thus, I would, in principle, support publication of this article with minor revisions (see minor points below).

Response: We thank the reviewer for their very positive assessment of our work.

My only concern is that a recent publication by Zhang et al. in *Molecular Cell* (doi: 10.1016/j.molcel.2023.03.015) already reported the cryo-EM structures of the pre-tRNA-bound human TSEN complex in the pre-catalytic state as well as in the post-catalytic state at resolutions of 2.94 and 2.88 Å, respectively. The pre-catalytic state structure by Zhang et al. and their mechanistic conclusions are very similar to those reported here by Yuan et al., raising the question whether Yuan et al. can provide further insight beyond what has already been published.

Thanks for pointing this out. Now we have included the comparison with the structures of Zhang et al., and point out the differences and similarities in both writings and as Supplementary Fig.12. The structure presented in this study provides unique insights lacking in their structures. To specify: 1) We captured the structure in pre-catalytic state by using the WT protein and slowing down the reaction at low temperature (please also see the response to central point below), which allows us to observe clear density of the scissile phosphate at this active site and get insights into the action of the cleavage reaction. 2) We captured the KRKRR motif contained N terminal extension of TSEN2 near the 5' splice site, which is important for the cleavage. Now we have added the mutational analysis to verify this importance (Figure 5d, lanes 3 and 4). Since the N terminal extension (NTE) of TSEN2 is missing in all other structural studies (including PDB entry 7UXA, 7ZRZ, 8HMY and 8HMZ), we speculate that the NTE of TSEN2 is flexible until it cooperatively traps G38 with wild type catalytic triad, thereby locking the scissile phosphate into the catalytic pocket. We have included related descriptions in Discussion (p.12). We believe these two points provide unique and important insights missing in other structures due to the different strategy in sample preparation.

A central point Yuan et al might address in more detail in a revision is the fact that the pre-catalytic state structure by Zhang et al. was prepared using enzymatically inactive catalytic site mutants. However, in this context, it would be important to clarify, whether the structure reported here is indeed pre-catalytic and that it does not represent a heterogenous mix of unprocessed and partially processed pre-tRNA in the post-catalytic state (e.g. present a detail of

the electron density around the 3' cleavage site which supports the claim that it is indeed the pre-catalytic state).

Thanks for this suggestion. We apologize for our oversight that the electron density of the cleavage sites was not included in the original manuscript. Now we have provided a detail of electron density around the 3' and 5' cleavage site, showing the connection of the phosphate group and ribose. Please note that the connection of the 5' cleavage site is missing in other structures (Supplementary Fig. 5a, 5c and 5f). We believe this continuous density strongly supports our claim that the structure is captured in a pre-catalytic state.

Further, in Zhang et al., the sample was incubated at 37 °C for 20 minutes, and the pre-tRNA had been almost completely cleaved based on their Urea-PAGE gels. In this work, we tried to capture in action structure of the wild-type complex by keeping the sample at 4 °C or on ice to slow down the reaction, which had been successfully tried in my previous work (Sun et al. DOI: 10.1126/science.aaz7758). Now we have added a cleavage assay (now Supplementary Fig. 1a) to show that most pre-tRNA was still intact after incubation on ice for 45 minutes, consistent with our structural finding.

Thus, based on the structural details and biochemical assay, we suggested the structure was captured in a pre-catalytic state.

Minor points:

- Please provide evidence for the claim that the reported structure represents a homogenous (pre-catalytic) state.

We have presented a detail of the electron density around the 3' cleavage site and the 5' cleavage site (see Supplementary Figs. 5a-c and 5f) for the claim that the reported structure represents a pre-catalytic state. And our further 3D classification didn't show different conformations. However, based on the structure from Zhang et al., the overall conformational change between pre-catalytic and post-catalytic state is subtle. We can't rule out the possibility that a small portion of pre-tRNA was cleaved and stay on the complex. It's technical difficult to classify out the connected or broken phosphodiester bond since the overall conformational change is subtle. But because the reported density map represent an average of the particles, the continuous density in the cleavage site suggests the majority of the particles are in the pre-catalytic state. This is also supported by our cleavage assay showing most of the tRNA are intact in our sample (now Supplementary Fig. 1a). Thus, our reported structure likely represents a homogenous pre-catalytic state.

- Line 36: “beautifully” seems a somewhat odd choice of wording in a scientific context.

We have deleted the word “beautifully” in Line 36.

- Lines 157 and 228: Does the structure provide a clue as to why “additional structural characteristics” in eukaryote TSEN are critical for pre-tRNA recognition, whereas they are dispensable in Archaea?

The pre-tRNA splicing machinery in archaeal and eukaryotic endonucleases is highly conserved. However, there have been evolutionary features in eukaryotes, resulting in several new insertions in the TSEN complex. Our structure found that certain “additional structural characteristics”, such as the NTE of TSEN54, the L α of TSEN34, and the C tail of TSEN2, cooperatively recognize mature body parts of pre-tRNA. These findings shed light on why eukaryote TSEN requires the presence of the mature domain of pre-tRNA for efficient cleavage, making it more effective in the selection of RNA substrates in vivo. In eukaryotes, tRNA introns are strictly located at the canonical position between nucleotides 37 and 38 of the mature tRNA, whereas intron positions in archaeal tRNAs are variable, including within the anticodon loop, the anticodon stem, the variable loop, the D- and T-arms, and the acceptor stem. Therefore, archaeal splicing endonucleases primarily recognize and cleave a distinct RNA architecture known as bulge-helix-bulge (BHB), which is independent of the mature body of the tRNA. The “additional structural characteristics” observed in eukaryotic TSEN are not necessary in Archaea. We have revised this part in Discussion (p.11): “...such as the NTE of TSEN54, the L α of TSEN34, and the C tail of TSEN2, cooperatively recognize specific regions of mature body of pre-tRNA. This sheds light on why eukaryotic TSEN requires the presence of the mature domain of pre-tRNA for efficient cleavage, ultimately enhancing its ability to effectively select RNA substrates in vivo.”

Reviewer 2:

In this manuscript, Yuan et al. present the cryo-EM structure of human tRNA splicing endonuclease (TSEN) complexed with an intron-containing pre-tRNA. The structure elucidates the organization of the four subunits within the intact assembly and their role in recognizing and cleaving the pre-tRNA. Notably, the presence of insertion loops originating from TSEN54 and TSEN2, which coordinate the splicing sites, adds an intriguing aspect. Additionally, a comparative analysis between human TSEN and its archaeal counterparts highlights the conserved structural features shared across kingdoms. The structural data and insights presented here overlap significantly with a recent work (Zhang et al. 2023, Molecular Cell), but contain unique findings. Overall, this study is of high quality and provides valuable insights into pre-tRNA splicing. However, several points need to be addressed before publication in Nature Communications.

Response: We thank the reviewer for their positive assessment of our work.

Major:

-page4, line 94: The authors state that a cryo-EM reconstruction was obtained using only 6% of the input particles. This raises the question of whether the samples are heterogeneous. However, the details of the data processing pipeline (fig. S2) are not clearly described. Specifically, the heterogenous refinement resulted in four 3D classes that appear highly similar. Are these classes representing distinct working states of the complex? It is recommended that the authors provide additional information regarding the data processing steps and elaborate on the criteria used for selecting particles for the reconstruction.

Thanks for the reviewer's suggestions. We have made modifications to our data processing pipeline (see Supplementary Fig. 2). After the heterogenous refinement, we observed that one of the four classes contain more densities belongs to TSEN2 (335K particles, depicted in purple). In order to enhance clarity, we have adopted a new view to show the map images of the four classes, which revealed additional density located at the bottom of the third class. The density of TSEN2 in the first two classes is relatively incomplete, while the rest parts of the complex appear similar. The quality of the fourth class (cyan) is poor. Upon assessing the integrity of TSEN2 and considering the overall quality of the map, we decided to focus on further processing the third class. We have also attempted to combine the first three classes to improve the map quality of the core, but the improvement is only subtle.

To address the reviewer's concern that only 6% of the particles was used for the final reconstruction, they are mainly because, 1) To avoid missing any possible particles, especially the top views with small projection size, and the possible sub-complexes, we used a low threshold to pick particles, which can lead to the picking several junk particles. 2) Our complex is in active form, though low temperature was used to slow down the reaction, it may still have different conformational states which are too dynamic to capture a defined structure. 3) The selection of the complete TSEN2 leads to another round of particle lose.

Now we have modified the image processing sessions in Methods to : "4500 image stacks were motion-corrected and dose-weighted in MotionCor2³⁷. The contrast transfer function (CTF) parameters were determined with CTFFIND4³⁸ implemented in RELION-3³⁹. 5,263,873 particles were automatically picked from 4,345 micrographs with Gautomatch (<https://www.mrc-lmb.cam.ac.uk/kzhang/Gautomatch/>) using the selected 2D averages from a small subset of the data as templates. The particles were extracted and normalized into 200x200 boxes. To avoid the loss of the particles in rare orientations, 2D classification step was skipped and the extracted particles were directly subjected to Relion 3D classification into six classes using the initial model obtained with Cryosparc⁴⁰ ab-initial reconstruction. Among the six classes, only one class showed clear structural feature of a complex, while others shows much less secondary structures or the complex is less complete. 1,568k particles in the good class were selected and subjected to another round of 3D classification in Cryosparc. One of the four classes contain more densities belongs to TSEN2 (335K particles, depicted in purple). The additional densities are labeled with a '*' (star). While the rest parts of the complex are similar with the first two classes. The quality of the fourth class (cyan) is poor. The alignment parameters are optimized in Cryosparc non-uniform refinement to generate a map at 3.44 Å. To further improve the map quality, these 335K particles were processed with 3D auto-refine, CTF refinement, and Bayesian polishing in RELION-3. After these processes, a map at an overall resolution of 3.19 Å was obtained with Cryosparc non-uniform refinement. To further improve the local resolution of "KRKRR" motif, a soft mask near the "KRKRR" region was applied during the Cryosparc Local Refinement."

-page6, line132-134: The authors propose that the nucleotides near the 5' SS may form a similar arrangement with 3' SS bulge. I not convinced by this analysis since the cryo-EM map quality in this region is rather poor (fig. S4). The author should provide clear EM density maps to support their structural findings. Indeed, there' s only one panel in fig. S4 that shows the EM map, which is insufficient. The authors should provide more details of the EM maps for the RNA conformation as well as the critical interactions described in their main text.

Thanks for the reviewer's suggestion. We have included the following density maps as Supplementary Figure 5:

Supplementary Fig. 5a: the 3' bulge with the A-I base pair, and 5f to show the pre-catalytic state. 5b and 5e: the R409-R452 tweezer for the first bulge nucleotide, C51 and the R279-W306 tweezer for the nucleotide, U36. Since the density for U36 is poor, we have included the mutational analysis to verify their critical role in forming the correct conformation of the 5' splice site (Figure. 5d, lane5 and Supplementary Fig.10b). The double mutants R409A and R452A exhibited a slower cleavage rate, leading to the accumulation of the 3'-exon-intron intermediate. In contrast, the mutation of the R279-W306 tweezer led to the complete abolishment of 5' splice site cleavage (Figure. 4c and 5d). We have added related discussion on this difference in main text (p.12): "our mutagenesis and biochemical experiments have provided further insights into the functional roles of specific residues in TSEN. We found that double mutants of the R279-W306 tweezer are lethal for the 5' splice site, whereas mutations of the R409-R452 tweezer retain activity for the 3' splice site, indicating that the correct conformation of the 5' splice site is strictly dependent on the combined contributions of both the R279-W306 tweezer"

5c and 5f: the scissile phosphates at the active site with the catalytic triad residues for 3' SS and 5'SS.

-page6, line 136-141: The authors discussed the recognition of the mature body of pre-tRNA in this paragraph. However, the figures presented here are of low quality, making it challenging to identify the specific interactions mentioned in the text. To support their findings, it is crucial for the authors to provide the EM maps that justify their conclusions. Furthermore, model validations, model-to-map FSC curves, should also be included during revision.

Thanks for this suggestion. We have provided the EM maps of the interface between TSEN and pre-tRNA as supplementary Figure. 6. Related to Figures 3b, 3c, and 3f.

We have also included the model validations in Supplementary Table 1 and model-to-map FSC curves as Supplementary Fig. 2e.

-page8, line 194-216: This paragraph describes the conformation of 5' SS and the interactions that contributes to its recognition. However, the poor EM density maps shown in fig. S4 does not support the modeling of 5' SS. Indeed, there's almost no discernible EM density for U36-G38. We are also not shown by any EM density maps for the critical residues from TSEN2, including the "KRKRR" motif, which are implicated in 5'SS recognition. To justify their model, the author may improve the local resolution of this region and provide the EM density maps in the supplementary information. Otherwise, the description on 5' SS is purely speculative and should be removed from the text.

We have included the density maps of 5'SS as Supplementary Figs.5e and 5f, which shows U36 with the R279-W306 tweezer and G38 with the catalytic triad residues of TSEN2. We have included the mutational analysis to verify their critical role (Fig.5d and Supplementary Fig.10b). The mutation of the R279-W306 tweezer in TSEN34 or the mutation of the catalytic triad residues of TSEN2 led to the complete abolishment of cleavage at the 5' splice site. Mutating R73A and K361A in TSEN2 slowed down the cleavage rate.

To further improve the local resolution of “KRKRR” motif, a soft mask near the “KRKRR” region was applied during the Cryosparc Local Refinement (see supplementary Figure. 2e). However, we did not observe significant improvement (see the image below).

We have included the density map of the NTE of TSEN2 as Supplementary Fig.5g, and we found that the cleavage activity of TSEN was decreased by either deleting the KRKRR motif of TSEN2, or mutating it to AAAAA (result in the accumulation of the 5'-exon-intron intermediate and less 5' exon products, see Fig.5d), supporting its important role in stabilizing the negatively charged intron.

Minor:

-page4, line71: Reference is required here.

I have added one related reference “C. R. Trotta, S. V. Paushkin, M. Patel, H. Li, S. W. Peltz, Cleavage of pre-tRNAs by the splicing endonuclease requires a composite active site. Nature 441, 375- 377 (2006)” here.

-page6, line 128: “Cryo-EM” should be “cryo-EM”

I have changed “Cryo-EM” to “cryo-EM”.

Reviewer 3

Yuan et al. report RNA recognition and cleavage mechanism during tRNA splicing mediated by human TSEN endonuclease based on a single cryoEM structure. Though this work is less thorough than that recently published by Zhang et al. (<https://doi.org/10.1016/j.molcel.2023.03.015>) or other publications in press of the same complex (PDB ID: 7uxa for instance), it provides some unique information lacking in these other structures. However, it should address the following concerns.

1) It is possible that authors did not realize the publication by Zhang et al. (PDB id: 8hmz and

8hmy) or the structure 7uxa. However, since all these structures are nearly identical with slightly different constructs and catalytic mutations or with CLP1, authors should provide a comparison and point out the differences and similarities in both writings and as supplementary figures. For instance, 8hmz is also active form but its 3' splice site is cleaved and modeled with the product. But in this work, the 3' splice site is intact. Can authors address why this is the case?

Thank you for this suggestion. We have provided a comparison (now Supplementary Fig. 12) and added a new related paragraph in the Discussion section (p. 12): "Another cryo-EM structure of the human TSEN bound to pre-tRNA was recently published³⁴, as well as other publications of the same complex (PDB ID: 7UXA and PDB ID: 7ZRZ)^{35,36}. These studies have provided"

For study on CLP1, we have revised this sentence in Discussion (p. 13): "These regions may recruit other regulatory factors for metazoan tRNA processing, Clp1 for instance." and cited the Zhang et al reference and Simon et al reference.

For the 3'SS with wt TSEN, Zhang et al. and our work captured the pre-tRNA in different states. In Zhang et al., the sample was incubated at 37 °C for 20 minutes, and the pre-tRNA had been almost completely cleaved based on their Urea-PAGE gels. In this work, we tried to capture in action structure of the wild-type complex by keeping the sample at 4 °C or on ice to slow down the reaction, which had been successfully tried in my previous work (Sun et al. DOI: 10.1126/science.aaz7758). We have added a cleavage assay (now Supplementary Fig. 1a) to show that most pre-tRNA was still intact after incubation on ice for 45 minutes and revised this sentence in the main text (p. 4): "...and kept the sample at 4 °C or on ice, which preserved the integrity of most pre-tRNA molecules during the process of freezing the grid." We have also included the following in Discussion (p. 13): "based on the structure in the post-catalytic state, the nucleotides in the 3' bulge and the A-I stem remain stably associated with TSEN. It might be not excluded that there is few 3' cleaved product present in our sample that cannot be distinguished by 3D classification."

Interestingly, authors captured the N terminal extension of TSEN2 near the 5' splice site while other structures did not, thereby providing some unique insights. Thus the comparison should help to emphasize the common conclusions while highlight unique contributions.

We thank the reviewer's insightful suggestion. For the 5'SS, based on our biochemical assay, we can only observe the accumulation of the 5'-exon-intron intermediate, but no 3'-exon-intron intermediate in WT samples (Fig.4c lane2; Fig.5d lane2), which consistent with the cleavage assays from Zhang et al.. It suggests that the differences between cleavage rates at the 3' and 5' splice sites, and cleavage at 5'SS is slower. The importance of the N terminal extension of TSEN2, especially the KRKRR motif has been demonstrated in the mutagenesis and biochemical experiments (now Figure. 5d). This unique information supports that our structure is in a different state from all the other structures. We have added the following sentences into Discussion (p. 12): "Interestingly, the complex structure presented in this study differed from both the catalytic-dead state and the post-catalytic state. This is supported by the presence of the NTE loop of TSEN2 (Supplementary Fig.9), a critical feature missing in all other structural studies. It is speculated that the NTE of TSEN2 is flexible in structure until it cooperatively traps

G38 with the catalytic triad of residues in TSEN2, thereby locking the scissile phosphate into the catalytic pocket.....”

Another difference between our structure and that of Zhang et al. is the assignment of A-I stem. The density surrounding A-I stem exhibits good quality, and there was no indication of any density being pulled out or disconnected from the A-I stem, which is actually the same in all the maps for this complex. In contrast, Zhang et al. reported a pulled-out U33 in their structure. As we show, TSEN recognize the mature conformation of pre-tRNA, effectively acting as a molecular ruler. The positions of the catalytic pockets on TSEN are well-defined, and thus, the length of A-I stem and the assignment of nucleotides from U33 to G38 are quite important, which will influence the distance between the active site of TSEN2 and the scissile phosphate of G38. We have provided a detail of the electron density around the A-I stem in Supplementary Fig.5d.

2) Various structures captured partial 5' splice site and none seems to show correct positioning of the scissile phosphate in the predicted active site. This is also a good region where authors can compare and comment. Did their structure show correct placement of 5' scissile phosphate and if so, do active site residues cooperate with the location? If not, what is the likely reason?

Thank you for this suggestion. According to our study, the catalytic triad residues of TSEN2 (Y369, H377, and K416) play crucial roles in the cleavage of the 5' splice site, which is consistent with the mutational analysis conducted by Zhang et al. (<https://doi.org/10.1016/j.molcel.2023.03.015>). However, the structures presented in their study, including the pre-catalytic state (PDB ID: 8HMY, with the H377A mutant) and the post-catalytic state (PDB ID: 8HMZ, WT), as well as other structures (PDB ID: 7UXA, catalytic triad residues mutated to Ala; PDB ID: 7ZRZ, with the H377F mutant), do not directly elucidate the importance of the catalytic triad (K416 for instance), as they fail to observe tRNA intron density at the 5' splice site. In contrast, we observed clear density of the scissile phosphate at this active site, and the base of G38 was exactly π -stacked with H377 (see Supplementary Fig.5f). We compared our structure with other four recently released PDB structures.

The model comparison is now showing as Supplementary Fig. 12, and the related legend has been updated. Additionally, we have added a new paragraph in the Discussion section (p. 12): “.....Notably, the conformations of the 5' splice site in different structures varied. In the case of mutating all three catalytic residues to Ala in TSEN2, no RNA density was observed in the binding pocket of the 5' splice site (Supplementary Fig.12a and b). The crucial residue K416 exhibited a different conformation, being distant from the active site when using truncated TSEN with the H377F mutation (Supplementary Fig.12c). Similarly, in the TSEN complex with TSEN2 H377A mutant, the distance between the active site and the scissile phosphate of G38 is too far (Supplementary Fig.12d). For the structure of wild-type TSEN in the post-catalytic state, the sample revealed almost complete cleavage of the products, resulting in the absence of density for the scissile phosphate (Supplementary Fig.12e)³⁴. Interestingly, the complex structure presented in this study differed from both the catalytic-dead state and the post-catalytic state. This is supported by the presence of G38 and the NTE loop of TSEN2 (Supplementary Fig.9), a critical feature missing in all other structural studies. It is speculated that the NTE of TSEN2 is flexible in structure until it cooperatively traps G38 with the catalytic triad of residues in TSEN2,

thereby locking the scissile phosphate into the catalytic pocket. All these studies collectively provide compelling evidence that any catalytic mutation in TSEN2 not only abolishes the cleavage activity but also weakens the binding affinity between TSEN2 and pre-tRNA,.....”

3) The work is mostly structural with almost no functional verification. Given that some of the findings are unique, it would be useful to verify structural findings by mutagenesis. For instance, the lysine residues one TSEN54 near 3' splice site (K41) and that of TSEN2 (K12, or KRKRR) seem to play quite an important role in possibly stabilizing the negatively charged intermediate. Mutational analysis would be very helpful.

Thank you for this suggestion. We have included the mutational analysis to verify our structural findings. Our mutagenesis and biochemical experiments support the structural observations.

For various mutants on 3'SS cleavage, we found that the K41A mutation in TSEN54 has minimal impact on its own. However, K41A can increase the impact of R409A (TSEN2) and K239A (TSEN34) double mutants. We have added the following sentences in main text (p. 8): “Our structural observations are supported by the mutational analysis (Fig. 4c). The cleavage activity for the 3' splice site was completely abolished by mutating the catalytic triad of residues to Ala in TSEN34 (lane 6). While the R409A mutation alone had minimal impact, the double mutants R409A and R452A, which affect the cation- π recognition of the first bulge base, exhibited a slower cleavage rate, leading to the accumulation of the 3'-exon-intron intermediate (lanes 3 and 4). Interestingly, the residues R409A (TSEN2), K239A (TSEN34), and K41A (TSEN54) are involved in capturing the ribose-phosphate backbone in the catalytic pocket. We found that introducing the K41A mutation in TSEN54 or the R409A mutation in TSEN2 enhances the effect of the double mutants formed by R409^{TSEN2}/K239^{TSEN34} or K41^{TSEN54}/K239^{TSEN34}, resulting in a complete loss of cleavage. This observation strongly suggests that the residues R409^{TSEN2}, K239^{TSEN34}, and K41^{TSEN54} play crucial roles in stabilizing the conformation of the 3' bulge. (Fig. 4c, lane 5 and Supplementary Fig.10a).”

For various mutants on 5'SS cleavage, we found that the cleavage activity of TSEN was decreased by either deleting the KRKRR motif of TSEN2, or mutating it to AAAAA. We have added the following sentences in main text (p. 10): “To verify our structural findings, we carried out mutagenesis and biochemical experiments (Fig. 5d). In comparison to the wild-type TSEN, the mutation of the catalytic triad of residues in TSEN2 abolished the cleavage activity at the 5' splice site (lane 7). Notably, deleting the KRKRR motif of TSEN2, or mutating it to AAAAA greatly reduced the cleavage activity at the 5' splice site, resulting in the accumulation of the 5'-exon-intron intermediate and fewer 5' exon products (lanes 3 and 4). This observation supports the important role of the KRKRR motif in stabilizing the negatively charged intron. Remarkably, despite the poor density of U36 and A37, the mutation of the R279-W306 tweezer in TSEN34 led to the complete abolishment of cleavage at the 5' splice site (lane 5; Supplementary Fig.10b), which differs from the mild impact of the mutation of the R409-R452 tweezer at the 3' splice site (Fig. 4c, lane 3). Furthermore, mutating R73A and K361A residues in TSEN2 significantly slowed down the cleavage rate, indicating their potential involvement in transiently stabilizing the phosphate backbone of A37 during cleavage (lane 6). Hence, our biochemical studies provide support for the structural observations. Similar to the 3' splice site, the residues for the recognition of the first bulge base (R279-W306 tweezer), the stabilization of the ribose-

phosphate backbone (R73 and K361), and the cleavage process (the catalytic triad) are important for the 5' splice site. In addition, the KRKRR motif in the N-terminal insertion of TSEN2 is unexpectedly required for highly efficient cleavage. These findings from the mutational analysis enhance the overall understanding of the mechanisms involved in the cleavage process of the 5' splice site."

The mutational analysis is now showing as Figure. 4c (for 3' SS) and Figure. 5d (for 5' SS), and the related legend has been updated.

4) In the intact 3' splice site that is not cleaved, do authors see the proposed inline geometry? If so, please point it out and mark it so.

We have incorporated an insertion in Figure 4b to show the proposed inline geometry. Additionally, we have included the following sentence in the main text (p.8): 'The in-line geometry of the attacking 2'-oxygen, the scissile phosphate, and the leaving 5'-oxygen is formed (Fig. 4b, insert).' The legend accompanying this panel now states: 'The insert presents the in-line geometry created by the attacking 2'-nucleophile oxygen, scissile phosphate, and the leaving 5'-oxygen.'

5) Line 55, please revise the statement "In archaeal EndA, the complex is composed of four subunits that form different architectures, including α_4 , α'_2 , $(\alpha\beta)_2$, or ε_2 -type complexes(11, 12)," In fact, the endonucleases have the same architecture that is arranged by using different domains and subunit composition. Some do not have four subunits.

We have revised this sentence in main text (p. 3): 'In archaea, four distinct types of EndAs that share similar architectural arrangements but employ different domains and subunit compositions are observed, including α_4 , α'_2 , $(\alpha\beta)_2$, and ε_2 -type complexes.'

6) Please specify which restriction enzyme was used to linearize the pre-tRNA-encoding plasmid for transcription;

We did not use any restriction enzyme to linearize the pre-tRNA-encoding plasmid. Instead, DNA templates for transcription were obtained through PCR amplification. We have revised this part in the Methods: "DNA templates for transcription were generated using PCR and then extracted by PEG6000. The 5' primer sequence used was TAATACGACTCACTATAgGGGGC, with the T7 promoter sequence underlined. Pre-tRNAs were amplified *in vitro* using the T7 High Yield RNA Transcription kit (Vazyme). The site of transcription initiation is indicated in lower case, resulting in the addition of the sequence "GGG" at the 5' -end of pre-tRNA^{Arg-TCT}, thereby increasing its length to 91 nt."

7) The rotamer outlier content is high for this resolution, so is core clash and Ramachandran outliers. Please fix them.

We have fixed the model and included new statistics in Supplementary Table 1. Now the rotamer outlier is 0.39%.

8) Report min/mean/max B factors and model FSC resolutions

We have included min/mean/max B factors in Supplementary Table 1 and model-to-map FSC curves as Supplementary Fig. 2e.

9) Please label the axes of the FSC plot;

We have labeled the axes of the FSC plot, now Supplementary Fig. 2d.

10) Please label the unit for the local resolute plot color bar

We have added the unit 'Å' for the local resolute plot color bar, now Supplementary Fig. 2f.

Reviewers' Comments:

Reviewer #1:

Remarks to the Author:

2. Review for Yuan et al.

In their revised manuscript "Recognition and cleavage mechanism of intron-containing pre-tRNA by human TSEN endonuclease complex", Yuan et al. addressed and answered all points raised during the initial review. Their changes significantly improved the quality and presentation of their work and now includes the requested discussion of other recent studies reporting cryo-EM structures of the TSEN complex.

There are a few minor points, which the authors should address before publication:

1. There are still a number of typos/grammatical issues throughout the text, which should be edited carefully (e.g. lines 92/93: "This way helped minimized the cleavage reaction...")
2. Similarly, the wording should be checked - e.g. the description of mutations as "lethal" (line 306) is odd.
3. In Figures 4c and 5d, the authors claim that the cleavage assays were repeated three times. However, they only show a single representative result, which defeats the objective to provide evidence for reproducibility. To actually demonstrate the reproducibility of their results, the authors should include these other repetitions of the cleavage assays at least in the supplementary information (e.g. as a summary in form of a densitometric analysis of all their gels).

Reviewer #2:

Remarks to the Author:

I think the authors have improved their manuscript following the reviewer comments. However, one major concern remains, which must be addressed prior to publication.

In the first round of review, concerns were raised regarding the model building of the 5'SS due to the lack of cryo-EM density for U36-A37-G38. In the revised version, the authors provided a supplemental figure (Fig. S5) to display the cryo-EM density maps. However, upon examining Figs. S4 and S5, I remain unconvinced by the proposed assignments. The cryo-EM density maps for U36 and G38 exhibit low quality and do not provide sufficient information for model building. Additionally, the density map for A37 is completely invisible in their cryo-EM reconstruction. The poor density observed could be resulted from either sample heterogeneity or intrinsic flexibility of the 5'SS. Overall, I think this study provides mechanistic insights into the recognition of 3'SS. But for the 5'SS, I think the authors have over-interpreted their data. The local cryo-EM density map shown in Figs. S4 & S5 does NOT support for the modeling of 5'SS.

Reviewer #3:

Remarks to the Author:

Authors now addressed my concerns and added significant amounts of mutagenesis analysis, which has improved the manuscript significantly. There are some minor concerns raised as a result of the mutagenesis work:

Page 12, Line 286, The meaning of "In the case of mutating all three catalytic residues to Ala in TSEN2, no RNA density was observed in the binding pocket of the 5' splice site (Supplementary Fig.12 . a and b)." is not clear. If this refers to other structures, please add citations in addition to referring to Figure S12;

Authors cloned and purified a number of mutants for cleavage assays. Since it is not known how

mutations impact the integrity of the enzyme, it is necessary to show at least an SDS-PAGE gel of all enzymes used for the assay.

Please comment on why Trotta et al. (DOI: 10.1038/nature04741) showed the cation- π is not important for 3' splice site whereas the data presented here clearly indicate its importance. Did Trotta et al. incorrectly predicted the cation- π residues of Sen2?

Responses to reviewers' comments (our responses are in black)

Reviewer 1:

In their revised manuscript “Recognition and cleavage mechanism of intron-containing pre-tRNA by human TSEN endonuclease complex”, Yuan et al. addressed and answered all points raised during the initial review. Their changes significantly improved the quality and presentation of their work and now includes the requested discussion of other recent studies reporting cryo-EM structures of the TSEN complex.

Response: We thank the reviewer for the positive comments of our work.

There are a few minor points, which the authors should address before publication:

1. There are still a number of typos/grammatical issues throughout the text, which should be edited carefully (e.g. lines 92/93: “This way helped minimized the cleavage reaction...”)

Thanks for pointing this out. We apologize for the typos/grammatical issues. We have made several necessary corrections, and shown them in red words in the revised manuscript, including lines 92/93, which now read: "This way helped minimize the cleavage reaction..."

2. Similarly, the wording should be checked - e.g. the description of mutations as “lethal” (line 306) is odd.

We appreciate the reviewer's thorough reading. We have removed the word “lethal” and revised this sentence in main text (p. 13): “...double mutants of the R279-W306 tweezer completely abolished cleavage at the 5' splice site...”

3. In Figures 4c and 5d, the authors claim that the cleavage assays were repeated three times. However, they only show a single representative result, which defeats the objective to provide evidence for reproducibility. To actually demonstrate the reproducibility of their results, the authors should include these other repetitions of the cleavage assays at least in the supplementary information (e.g. as a summary in form of a densitometric analysis of all their gels).

Thanks for your suggestion. We have included related repetitions of the cleavage assays as Supplementary Fig. 10, b and c.

Reviewer 2:

I think the authors have improved their manuscript following the reviewer comments. However, one major concern remains, which must be addressed prior to publication.

In the first round of review, concerns were raised regarding the model building of the 5'SS due to the lack of cryo-EM density for U36-A37-G38. In the revised version, the authors provided a supplemental figure (Fig. S5) to display the cryo-EM density maps. However, upon examining Figs. S4 and S5, I remain unconvinced by the proposed assignments. The cryo-EM density maps for U36 and G38 exhibit low quality and do not provide sufficient information for model building. Additionally, the density map for A37 is completely invisible in their cryo-EM reconstruction. The poor density observed could be resulted from either sample heterogeneity or intrinsic flexibility of the 5'SS. Overall, I think this study provides mechanistic insights into the recognition of 3' SS. But for the 5' SS, I think the authors have over-interpreted their data. The local cryo-EM density map shown in Figs. S4 & S5 does NOT support for the modeling of 5' SS.

Thanks for your comments. We have now included the density maps of G38 from two different views for reference (Supplementary Fig. 5, f and g). The density of G38 is strong and continuous, not only located in the catalytic pocket of TSEN2 but also π -stacked with H377. In Supplementary Fig. 5f, we show that the scissile phosphate is located at the center of the catalytic triad of residues in TSEN2, Y369, H377, and K416, similar to the A53 in panel c. In Supplementary Fig. 5g, we aim to show the π -stacking interaction between H377 and G38.

In response to your concerned, we have reworded and added a description of how we performed the model building of the 5'SS as follow (p. 9-10, highlighted in yellow): "At the 5' splicing boundary, we observed two nucleotide densities stacking with H377 and W306, respectively (Supplementary Fig. 5, e-g). Due to the disrupted densities with neighboring nucleotides, we are unable to definitively assign these two nucleotides solely based on the densities. However, when overlaying TSEN34 and TSEN2, which share similar architecture, we observe an overlap of positions for both the key residues in the enzymes and also the RNA, as well as the cover loops, K41 in TSEN54 and KRKRR in TSEN2 (Fig. 5a). Thus, the interaction network at the 5' splice site in our structure exhibits high similarity to that of the 3' splice site. Given that G38 is the cleavage site, it is likely that the nucleotide stacking with H377 corresponds to G38, which also brings the one stacking with W306 assigned to U36. The strong density of the scissile phosphate indicates that the 5' splice boundary is also ready for cleavage (Supplementary Fig. 5, f and g). The scissile phosphate is located at the center of the catalytic triad of residues in TSEN2, Y369, H377, and K416....."

We have also added the sentence in discussion (p. 13) as follow: "Following studies are needed to further confirm the sequence assignment at the 5' splice site with optimized sample preparation and image processing protocols."

Reviewer 3

Authors now addressed my concerns and added significant amounts of mutagenesis analysis, which has improved the manuscript significantly. There are some minor concerns raised as a result of the mutagenesis work:

Response: We thank the reviewer for the positive comments of our work.

Page 12, Line 286, The meaning of “In the case of mutating all three catalytic residues to Ala in TSEN2, no RNA density was observed in the binding pocket of the 5' splice site (Supplementary Fig.12, a and b).” is not clear. If this refers to other structures, please add citations in addition to referring to Figure S12;

We have added the citation “Hayne, C. K. *et al.* Structural basis for pre-tRNA recognition and processing by the human tRNA splicing endonuclease complex. *Nat Struct Mol Biol*, doi:10.1038/s41594-023-00991-z (2023).” here, and we have added the PDB entry of the model with the mutations of all three catalytic residues to Ala in TSEN2 in the figure legend of Supplementary Fig.13a.

Authors cloned and purified a number of mutants for cleavage assays. Since it is not known how mutations impact the integrity of the enzyme, it is necessary to show at least an SDS-PAGE gel of all enzymes used for the assay.

Thanks for your suggestion. We have included a SDS-PAGE gel of all enzymes used for the assay as Supplementary Fig. 10a.

Please comment on why Trotta *et al.* (DOI: 10.1038/nature04741) showed the cation- π is not important for 3' splice site whereas the data presented here clearly indicate its importance. Did Trotta *et al.* incorrectly predicted the cation- π residues of Sen2?

Thanks for your suggestion. We have incorporated your suggested changes into the Discussion (p. 11) as follows: “...Interestingly, our findings appear contradictory to previous studies on the yeast Sen complex, where it was concluded that the cleavage of the 3' splice site is independent of the cation- π recognition by Sen2²². However, through sequence alignment and utilizing the AlphaFold model of yeast Sen2, we identified R321 and R369 of yeast Sen2 (instead of R321 and W348 in earlier biochemical studies²²) at positions corresponding to the arginine tweezer pair (R409 and R452) in human TSEN2. This finding suggests that the arginine tweezer pair at the 3' splice site is conserved in the yeast Sen complex as well....”

The figure below shows: Overlay of the structure of human TSEN2 reported here (in purple) with that of yeast Sen2 (light gray, AlphaFoldDB P16658). The arginine tweezer pairs and W348 are labelled.

R409, R452
R321, R369

Human TSEN2 (CTD domain) in this study
Yeast Sen2 (CTD domain) AlphaFold

Reviewers' Comments:

Reviewer #2:

Remarks to the Author:

The authors have addressed my concerns.

Reviewer #3:

Remarks to the Author:

I examined the provided coordinate and map. I concluded that

1) The 5' splice site model is indeed not supported by the map. The density for G38 suggests an alternative orientation 90 degree to the placed orientation. Given how weak the density is, though, authors should not place the model based on density. I suggest authors to acknowledge that they modeled 5' splice site mostly according to that in the archaeal structure, then support this model by the mutagenesis data. Perhaps this was how they originally intended.

2) There are many mistakes in the model. For instance, chain D, residue 63, is in a wrong orientation, which blocks the correct orientation of chain C, residue 20; the region of chain D, 34-38 is wrong; chain D residue 32 is also wrong, which blocks correct placement of chain E residue 23. Authors should go through the model residue-by-residue as it should have been done.

3) Since the resolution is not sufficient for authors to add hydrogen, authors should remove them from the model.

Responses to reviewers' comments (our responses are in black)

Reviewer #3

Response: We appreciate the reviewer's rigor regarding data quality and we thank the reviewer's precious feedback.

I examined the provided coordinate and map. I concluded that

1) The 5' splice site model is indeed not supported by the map. The density for G38 suggests an alternative orientation 90 degree to the placed orientation. Given how weak the density is, though, authors should not place the model based on density.

We appreciate your feedback regarding the density of G38. We agree with you that this density appears relatively weak, as depicted in Supplementary Fig. 5, f and g. This weak density may reflect the dynamic nature of the sample in an activating state. It underscores the inherent challenges associated with our research. Moreover, combined with the NTE loop of TSEN2 (a critical feature missing in all other structural studies), the complex structure presented in this study differs from both the catalytic-dead state and the post-catalytic state. Our findings significantly contribute to the field by offering fresh insights into the activation process. Therefore, I respectfully disagree with the reviewers comments that "authors should not place the model based on density" simply because the density is a little bit weak.

Regarding the orientation of G38, since there is some weaker density for U39 (which connects to the ribose of G38 and wasn't modeled) and the base of G38 that should form a pi stacking interaction with H377, I respectfully disagree with the reviewer's assertion that "The density for G38 suggests an alternative orientation 90 degrees to the placed orientation."

I suggest authors to acknowledge that they modeled 5' splice site mostly according to that in the archaeal structure, then support this model by the mutagenesis data. Perhaps this was how they originally intended.

We sincerely appreciate the reviewer's suggestion. However, it appears there might be a misunderstanding regarding our model-building process. Upon overlaying our model with the archaeal structure, as illustrated in Supplementary Figure 8, we were delighted to observe striking similarities in the recognition and catalytic mechanisms shared between human and archaeal RNA-splicing endonucleases. This exciting observation suggests the presence of a conserved mechanism in both archaeal and eukaryotic endonucleases. We would like to clarify that the archaeal structure served to further confirm our model-building assignment rather than the reverse order as mentioned by the reviewer.

In response to this comment, we have revised the sentence in the last paragraph of page 9 as follows: "Thus, the interaction network at the 5' splice site in our structure exhibits a high degree

of similarity to that of the 3' splice site and to that found in archaeal RNA-splicing endonucleases."

2) There are many mistakes in the model. For instance, chain D, residue 63, is in a wrong orientation, which blocks the correct orientation of chain C, residue 20; the region of chain D, 34-38 is wrong; chain D residue 32 is also wrong, which blocks correct placement of chain E residue 23. Authors should go through the model residue-by-residue as it should have been done.

We are very sorry for our careless mistakes. We have carefully reviewed the model and have made the corrections. We have included new statistics in Supplementary Table 1.

3) Since the resolution is not sufficient for authors to add hydrogen, authors should remove them from the model.

Thanks for pointing this out. We have remove the hydrogens from the model.